**Article** https://doi.org/10.1038/s41467-023-37368-1

# Post-acute sequelae of COVID-19 is characterized by diminished peripheral CD8+β7 integrin+ T cells and anti-SARS-CoV-2 IgA response

André Santa Cruz [1,2,3,4,7] ✉, Ana Mendes-Frias[1,2,7], Marne Azarias-da-Silva [5], Sónia André[5], Ana Isabel Oliveira[3], Olga Pires[3], Marta Mendes[3], Bárbara Oliveira [3], Marta Braga[3], Joana Rita Lopes [3], Rui Domingues[3], Ricardo Costa[3], Luís Neves Silva [3], Ana Rita Matos[3], Cristina Ângela[3], Patrício Costa [1,2], Alexandre Carvalho[1,2,3,4], Carlos Capela[1,2,3,4], Jorge Pedrosa[1,2], António Gil Castro[1,2], Jérôme Estaquier [5,6] ✉ & Ricardo Silvestre [1,2] ✉

Several millions of individuals are estimated to develop post-acute sequelae SARS-CoV-2 condition (PASC) that persists for months after infection. Here we evaluate the immune response in convalescent individuals with PASC compared to convalescent asymptomatic and uninfected participants, six months following their COVID-19 diagnosis. Both convalescent asymptomatic and PASC cases are characterised by higher CD8+ T cell percentages, however, the proportion of blood CD8+ T cells expressing the mucosal homing receptor β7 is low in PASC patients. CD8 T cells show increased expression of PD-1, perforin and granzyme B in PASC, and the plasma levels of type I and type III (mucosal) interferons are elevated. The humoral response is characterized by higher levels of IgA against the N and S viral proteins, particularly in those individuals who had severe acute disease. Our results also show that consistently elevated levels of IL-6, IL-8/CXCL8 and IP-10/CXCL10 during acute disease increase the risk to develop PASC. In summary, our study indicates that PASC is defined by persisting immunological dysfunction as late as six months following SARS-CoV-2 infection, including alterations in mucosal immune parameters, redistribution of mucosal CD8+β7Integrin+ T cells and IgA, indicative of potential viral persistence and mucosal involvement in the etiopathology of PASC.

COVID-19, regarded in early 2020 as an infectious disease with high virulence and mortality, also rapidly became recognized as a disease-causing important morbidity, as a significant number of patients reported persistent symptoms during convalescence[1]. This phenomenon was not new, as it had already been described in other viral infections caused by Influenza or Chikungunya and more recently by other members of the coronaviruses family as SARS-CoV-1 and MERS-CoV, which are similar in structure and genome to SARS-CoV-2[2–6].

Due to broad heterogeneity in terms of population, methods, timing and type of assessment, a global but precise estimation of the percentage of patients affected long-term by SARS-CoV-2 is not attainable. Yet, it is estimated that about half of hospitalized COVID-19

patients (31%–69%), and 10% of all patients, present post-acute sequelae of COVID-19 (also known as post-COVID-19 condition)[1,7–19]. Considering that 500 million people have been infected worldwide[20], the burden of COVID-19 sequelae is another giant face of the pandemic, with impact on individuals' quality of life, working capacity or autonomy and on healthcare systems[17,21–24].

As defined by the World Health Organization (WHO) in October 2021, post-acute sequelae SARS-CoV-2 condition (PASC) occurs in patients with a history of SARS-CoV-2 infection, at least three months after COVID-19 onset, with symptoms that cannot be explained by an alternative diagnosis and that last for at least two months with impact on patient functionality[25,26]. Despite there is no minimum number of symptoms, PASC is a multi-system disease, with a wide variety of unspecific physical and mental symptoms, that may vary in severity from mild to incapacitating[5,11,17,27–36]. Symptoms may have an onset after recovery from acute phase or may persist, fluctuate or even relapse[25]. The diagnosis is based mainly on the patient description.

Doubts remain about which patients are at increased risk for PASC development. Despite the existence of conflicting results, it seems that increased age and hospitalization (especially in Intensive Care Units) are risk factors for the persistence of symptoms[11,18,37–44]. Still, young, and non-hospitalized patients may also develop PASC[45]. Some studies also claim that women are more prone to specific manifestations of this syndrome[15,42,46]. Altogether, these results preclude the development of a predictive tool to closely monitor and treat the patients at risk after hospital discharge, a tool that would be of great value to patients and to the management and planning of healthcare systems' resources[47].

Several pathophysiological hypotheses have been proposed to explain the onset of PASC. Firstly, the extensive damage induced during acute disease by SARS-CoV-2 (considering the expression of ACE2, the SARS-CoV-2 receptor, at the surface of a myriad of epithelial and endothelial cells), may drive long-term tissue repair[5,36,48]. Secondly, the persistence of SARS-CoV-2 in human body, particularly in the gastrointestinal system, nervous system and other ACE2-expressing tissues, has been widely documented, and may remain for more than four months after acute infection[49–54]. The continuous viral replication could impact immune cell responses contributing to local immune activation and inflammation[52,53]. Thirdly, autoimmune phenomena have been reported in long-term recovered patients[55–59]. Thus, chronic immune activation due to virus persistence, autoimmunity, repair of damaged tissues or simply due to inability to downgrade acute inflammation, has been linked to PASC[30,60,61].

In this study, we follow the hypothesis that chronic immune dysregulation could characterise PASC. To address this issue, we perform both CD4[+] and CD8 + T cell immunophenotyping, quantify the viral-specific antibody response and determine the cytokine signature on groups of convalescent individuals who developed or not PASC. An immune CD8[+] T-cell activation is observed in convalescent patients, irrespectively of developing PASC in which the CD4/CD8 ratio remain low upon six months of infection. We also observe that patients show distinct profiles in type I and mucosal type III IFN compared to uninfected individuals. We pinpoint higher levels of CD8[+] T cells in PASC patients expressing the transcriptional factor Eomes. Furthermore, CD8[+] T cells expressing the β7 mucosal homing receptor are low in the blood of PASC individuals. Consistent with mucosal immune response, we detect specific IgA directed to N and S proteins of SARS-Cov-2 in PASC individuals. Thus, our results highlight a model in which the persistence of viral antigens in mucosa alters mucosal immune response.

## Results

### Demographic and clinical characterization of the cohort

We followed 72 patients in the consultation that were recruited during their hospitalization (at admission). We collected blood from those patients during their acute disease and at long term. We also recruited 55 outpatients, from whom we had no blood samples of their acute disease, only clinical and laboratory data. Thirty-seven healthy controls could be included after verifying the exclusion criteria. Comparison of the three groups revealed no significant differences between the median age: HC = 65 years (39–85), non-PASC = 61 years (25-87) and PASC = 64 years (24-85); H = 2.161, $p = 0.339$, $\eta_H^2 = 0.001$. The demographic characteristics of our cohort are described in Table 1. No significant dependence was observed between each comorbidity and the presence or absence of PASC, despite most patients being male (67%) and a tendency for a higher percentage of women with PASC. The median time from acute disease onset to blood collection was 165 days, equal in both groups. In our cohort, 81% of the patients were hospitalized during acute disease. We did not observe any difference in PASC prevalence according to acute disease severity, the need for hospitalization, lymphocyte count or commonly used inflammation markers (Table 2). Fatigue (45 patients, 73%) and dyspnoea (41 patients, 66%) were the most common symptoms in patients diagnosed with PASC, while neurocognitive symptoms were present in 21 patients (34%). Miscellaneous symptoms were reported by 16 patients: anxiety (1), arthralgia (1), dysphonia (3), erectile dysfunction (1), hair loss (1), loss of appetite (1), muscle weakness (1), myalgia (3), palpitations (1), persistent cough (4), sadness (1) and thoracic pain (1). The laboratory characterization of the cohort, during the appointment, along with the statistical analysis of each parameter is detailed in Supplementary Table 1. Altogether these data show that comorbidities, clinic observations during hospitalisation or common parameters obtained at the appointment are not directly linked to the occurrence of PASC.

**Table 1 | Demographic characterization of the cohort**

| Parameter | HC (n = 37) | Non-PASC (n = 65) | PASC (n = 62) | Statistical Analysis* | |
|---|---|---|---|---|---|
| | | | | Effect size | p value |
| **Gender**, n (%) | | | | | |
| Women | 14 (37) | 17 (27) | 25 (40) | 0.138 | 0.210 |
| Men | 23 (63) | 48 (73) | 37 (60) | | |
| Age, median (range) | 65 (39–85) | 61 (25–87) | 64 (24–85) | 0.001 | 0.339 |
| **Body mass index**, n (%) | | | | | |
| Normal | 17 (46) | 32 (49) | 29 (47) | 0.050 | 0.934 |
| Overweight | 12 (32) | 18 (28) | 16 (26) | | |
| Obese | 8 (22) | 15 (23) | 17 (27) | | |
| **Major Comorbidities**, n (%) | | | | | |
| Diabetes mellitus | 9 (24) | 19 (29) | 15 (24) | 0.055 | 0.777 |
| High Blood Pressure | 23 (62) | 34 (52) | 35 (57) | 0.075 | 0.627 |
| Dyslipidemia | 20 (54) | 27 (42) | 32 (52) | 0.109 | 0.377 |
| Tobacco use | 8 (22) | 10 (15) | 8 (13) | 0.132 | 0.221 |
| Alcohol abuse | 4 (11) | 1 (2) | 3 (5) | 0.163 | 0.113 |
| Chronic Obstructive Pulmonary disease | 3 (8) | 2 (3) | 4 (7) | 0.090 | 0.514 |
| Asma | 3 (8) | 3 (5) | 6 (10) | 0.087 | 0.537 |
| Obstructive Sleep Apnea | – | 4 (6) | 7 (11) | 0.171 | 0.092 |
| Chronic Kidney Disease | 1 (3) | 3 (4) | 3 (5) | 0.057 | 0.763 |
| Previous Cerebrovascular Disease | 3 (8) | 4 (6) | 6 (10) | 0.077 | 0.750 |
| Full autonomy | 37 (100) | 64 (98) | 61 (98) | 0.060 | 0.744 |

* For categorical variables Chi-square test was used to access the dependence of variables; effect size measures (Phi or Cramer's V to 2 × 2 comparations or more, respectively) and p value are reported. For scale variables, Kruskal-Wallis test was employed; effect size measure ($\eta_H^2$) and p value are reported.

**Table 2 | Clinical observations during hospitalization of non-PASC and PASC patients**

| Parameter | Non-PASC (n = 65) | PASC (n = 62) | Statistical analysis* | |
|---|---|---|---|---|
| | | | Effect size | p value |
| **Hospital Admission**, n (%) | | | | |
| Not hospitalized | 12 (18) | 12 (19) | 0.011 | 0.898 |
| Hospitalized | 53 (82) | 50 (81) | | |
| **Worst Disease Stage**, n (%) | | | | |
| I | 7 (11) | 8 (13) | 0.055 | 0.944 |
| IIa | 7 (11) | 5 (8) | | |
| IIb | 38 (58) | 36 (58) | | |
| III | 13 (20) | 13 (21) | | |
| **Maximum ventilatory support**, n (%) | | | | |
| Mechanical Invasive Ventilation | 9 (14) | 8 (13) | 0.066 | 0.997 |
| Mechanical Non-invasive Ventilation | 4 (6) | 4 (7) | | |
| High-Flow Oxygen | 12 (19) | 10 (16) | | |
| Conventional Oxygen (high) | 9 (14) | 10 (16) | | |
| Conventional Oxygen (low) | 17 (26) | 17 (27) | | |
| Room-air but needing health care | 1 (1) | 1 (2) | | |
| Room-air not needing health care | 1 (1) | 0 | | |
| Not hospitalized | 12 (19) | 12 (19) | | |
| **Simplified disease severity**, n (%) | | | | |
| Mild (Non-hospitalized to conventional oxygen) | 40 (62) | 40 (65) | 0.031 | 0.728 |
| Severe (High flow Oxygen to mechanical ventilation) | 25 (38) | 22 (35) | | |
| **Period between** (days) | | | | |
| Symptoms and diagnosis | 4 (0–14) | 5 (0–16) | 0.140 | 0.115 |
| Symptoms and hospitalization | 7 (0–14) | 9 (0–16) | 0.110 | 0.271 |
| Symptoms and appointment | 165 (60–285) | 165 (78–281) | 0.012 | 0.341 |
| Length of stay in the Hospital | 10 (2–95) | 12 (3–60) | 0.024 | 0.551 |
| **Laboratory parameters**, mean (range) | | | | |
| Ratio $PaO_2/FiO_2$ at admission | 269.3 (65–371) | 274.4 (103–491) | 0.129 | 0.176 |
| Worst Ratio $PaO_2/FiO_2$ | 147.8 (50–5888) | 130.1 (46–749) | 0.021 | 0.840 |
| Lowest lymphocyte number during hospitalization (lymphocytes/µL) | 800 (200–1900) | 700 (200–3200) | 0.073 | 0.425 |
| Highest C-reactive protein during hospitalization (mg/dL) | 158.4 (9–477) | 149.9 (2–393) | 0.062 | 0.496 |
| Highest Ferritin level during hospitalization (ng/mL) | 1752 (8–6866) | 1343 (133–12291) | 0.153 | 0.141 |
| Highest D-dimer level during hospitalization (ng/mL) | 1356 (341–4400) | 1387 (261–4400) | 0.020 | 0.830 |

* For categorical variables Chi-square test was used to access the dependence of variables; effect size measures (Phi or Cramer's V to 2 × 2 comparations or more, respectively) and p-value are reported. For scale variables, Mann-Whitney test was employed; effect size measure (r) and p-value are reported.

## Post-COVID-19 individuals have persistent alteration in CD8+ T cells

To determine the impact of PASC on T cell immunity, we first analysed the relative percentages of CD4+ and CD8+ T cells within the T lymphocyte population six months after infection in both PASC and non-PASC compared to healthy controls (HC). The percentages of peripheral blood CD4+ T were not significantly different among all groups, irrespectively of persistent symptomatology (H = 3.87, $p = 0.14$, $\eta_H^2 = 0.012$, Fig. 1A). In opposition, CD8+ T cells were different among groups (H = 17.9, $p = 0.0001$, $\eta_H^2 = 0.099$), with an increased percentage observed in both convalescent individuals compared to HC ($p = 0.0004$ for both comparations, Fig. 1B). This is reflected in reduced CD4/CD8 T cell ratios (H = 18.3, $p = 0.0001$, $\eta_H^2 = 0.10$, Fig. 1C), with a decrease observed in non-PASC and PASC compared to HC ($p = 0.0002$ and $p = 0.0008$, respectively). However, nor for CD4+, CD8+ T cells or CD4/CD8 T cell ratios, no interaction was observed between the acute disease severity and PASC (F(1, 164) = 0.009,

$p = 0.92$, $\eta_p^2 < 0.0001$ for CD4+T cells, F(1, 164) = 0.22, $p = 0.64$, $\eta_p^2 = 0.002$ for CD8+ T cells and F(1, 164) = 0.01, $p = 0.91$, $\eta_p^2 < 0.0001$ for CD4/CD8 ratio).

We then assessed the capacity of T cells to respond to SARS-CoV-2 antigens. Peripheral blood mononuclear cells (PBMC) were stimulated with N and S viral peptides and IFN-γ production was measured by ELISA. Our results indicated that SARS-CoV-2 convalescents displayed specific T cell response at a distal time point to viral infection in comparison to uninfected individuals (Fig. 2). However, we observed that PASC patients' lymphocytes produced less IFN-γ than stimulated lymphocytes from non-PASC patients (U = 166, $p = 0.021$, r = 0.33 and U = 183, $p = 0.013$, r = 0.38, Fig. 2A, B, respectively). To further address specific CD8+ T cell activation, we analysed by flow cytometry, cell surface expression of the early activation marker CD69. Besides a global significance was reached comparing the three groups (H = 18.5, $p < 0.0001$, $\eta_H^2 = 0.09$) no significant differences were found upon stimulation with N peptides

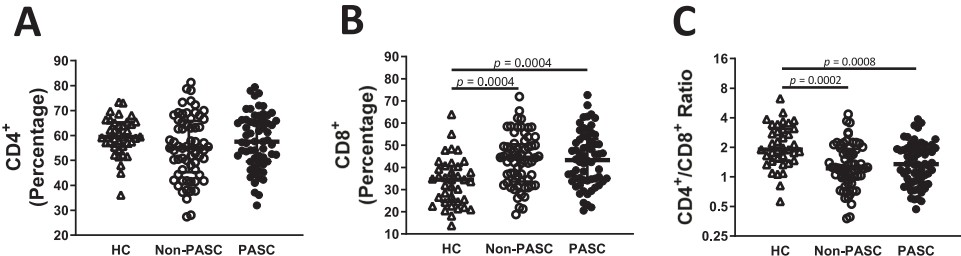

**Fig. 1 | Increasing levels of CD8⁺ T lymphocytes on convalescent COVID-19 patients.** Percentages of the CD4⁺ (**A**) and CD8⁺ (**B**) T lymphocytes populations based on gating strategy on Fig S1 (**B**) on healthy controls (HC; $n = 37$), non-presenting Post COVID-19 condition (non-PASC; $n = 65$) and presenting Post-COVID-19 condition (PASC; $n = 62$) individuals. CD4⁺/CD8⁺ ratio on HC, non-PASC and PASC individuals (**C**). Data are shown in a scatter dot plot format as median ± IQR; Kruskal–Wallis test was applied to identify statistical differences followed by Dunn's multiple comparisons test.

between non-PASC and PASC groups (Fig. 2C). Interestingly, when the activation was assessed upon stimulation with the S peptides (H = 50.8, $p < 0.0001$, $\eta_H^2 = 0.29$), a significant reduction was observed in PASC CD8⁺ T cells compared to non-PASC CD8 T cells ($p = 0.028$; Fig. 2D).

Thus, our results highlighted that compared to HC, convalescent patients six months after infection displayed higher levels of CD8⁺ T cells whereas CD8⁺ T cell immune response against S was lower in individuals with PASC.

### Convalescent patients display activated CD8 T cells

Because we observed higher levels of peripheral blood CD8⁺ T cells but lower S specific CD8⁺ T immune response in convalescent individuals presenting PASC symptoms, we assessed the levels of PD-1, LAG3 and TIM3, markers of T cell exhaustion[62]. Herein, we found significantly higher percentages of CD8⁺ T cells expressing PD-1 (H = 26.8, $p < 0.0001$, $\eta_H^2 = 0.16$, Fig. 3A), but not of CD4⁺ (H = 1.5, $p = 0.46$, $\eta_H^2 = 0.003$, Supplementary Figure 1A), specifically in convalescent individuals, whatever the presence of PASC symptoms status of individuals, when compared to HC ($p < 0.0001$ for both comparations). However, no significant differences were observed in the percentages of CD8⁺ or CD4⁺ expressing LAG3 or TIM3 in convalescent individuals when compared to HC (H = 2.2, $p = 0.32$, $\eta_H^2 = 0.002$ (Fig. 3B) and H = 3.9, $p = 0.14$, $\eta_H^2 = 0.014$ (Fig. 3C) for CD8⁺; H = 4.33, $p = 0.11$, $\eta_H^2 = 0.016$ (Supplementary Figure 1B) and H = 0.33, $p = 0.84$, $\eta_H^2 = 0.012$ for CD4⁺ (Supplementary Figure 1C)). These higher levels of PD-1 in convalescent individuals compared to HC could reflect exhausted T cells, although PD-1 can be expressed by T cells during activation and promoting effector memory CD8⁺ T cells[63–66].

Thus, considering the above-mentioned profile of CD8⁺ T cells and to clarify between activation and exhaustion, we looked at cardinal markers of the effector cytolytic T lymphocytes (CTL) program: perforin and granzyme A/B (GzmA and GzmB)[67]. T cells expressing PD-1 are associated with the loss of effector cytotoxic molecules[68]. Significant differences were found among groups in CD8⁺ T cells producing GzmA (H = 6.20, $p = 0.045$, $\eta_H^2 = 0.026$), GzmB (H = 10.53, $p = 0.005$, $\eta_H^2 = 0.055$) and Perforin (H = 6.36, $p = 0.042$, $\eta_H^2 = 0.028$) (Fig. 3D-F). Indeed, a significantly increased expression of GzmA, GzmB and perforin was observed in CD8⁺ T cells of PASC individuals compared to HC ($p = 0.043$ $p = 0.004$ and $p = 0.015$ for GzmA, GzmB and perforin, respectively; Fig. 3D–F). We also observed a significantly increased expression of GzmB in CD8⁺ T cells of non-PASC convalescent patients compared to HC ($p = 0.048$; Fig. 3E). To assess if acute disease severity had an impact in PASC, a two-way was performed to GzmA, GzmB and Perforin; however, no interaction was found among groups (F(1, 164) = 0.23, $p = 0.64$, $\eta_p^2 = 0.001$ for GzmA, F(1, 164) = 0.50, $p = 0.48$, $\eta_p^2 = 0.003$ for GzmB and F(1, 164) = 0.002, $p = 0.67$, $\eta_p^2 < 0.0001$ for Perforin, Fig. 3F). Thus, these results highlighted that CD8⁺ T cells from PASC individuals express higher levels of effector cytotoxic molecules.

We then assessed the expression of the T-box transcription factor 21 (T-bet) and eomesodermin (Eomes) factors that regulate maturation and effector functions of CD8 T cells. In particular, Eomes and PRF1 are highly expressed in memory CD8⁺ T cells[69–72]. While the percentage of T-bet expression in CD8⁺ T cells was similar among groups irrespectively of the presence or absence of PASC (H = 0.92, $p = 0.63$, $\eta_H^2 = 0.007$, Fig. 4A), Eomes presented significant differences among groups (H = 9.80, $p = 0.007$, $\eta_H^2 = 0.05$, Fig. 4B). In fact, the percentage of CD8⁺ Eomes⁺ was significantly increased in PASC patients when compared to HC and to non-PASC patients ($p = 0.017$ and $p = 0.031$ for HC and Non-PASC, respectively; Fig. 4B). Interestingly, a positive correlation was found between the percentages of CD8⁺ PD1⁺ and CD8⁺ Eomes⁺, which was exclusive of PASC individuals (r(62) = 0.24 and $p = 0.040$ for PASC and r(65) = 0.21 and $p = 0.14$ for non-PASC; Fig. 4C).

This increased expression of perforin and Eomes in CD8⁺ T cells from PASC individuals concomitant to higher expression levels of PD-1 indicates that CD8⁺ T cells are activated instead of being exhausted in individuals presenting PASC.

### Convalescent individuals with symptoms display higher levels of interferons, while β7⁺CD8 T cells are diminished in the blood

This chronic immune activation of CD8 T cells could be indicative of viral persistence in PASC patients[73–78]. Because detection of SARS-CoV-2 in blood is not a trivial issue, we evaluated the levels of type I and III interferons in the plasma of convalescent individuals as a readout of viral persistence. Whereas type I can be a marker of systemic innate immune activation, type III is related to mucosal compartment[79]. Regarding type I interferon, we observed differences in IFN-α2 levels among groups (H = 12.21, $p = 0.002$, $\eta_H^2 = 0.063$), specifically a significant reduction of IFN-α2 in PASC patients comparing to both non-PASC and the HC ($p = 0.004$ for both comparations; Fig. 5A). Of interest, this decrease was dependent on the acute disease severity (F(1, 80) = 4.52, $p = 0.037$, $\eta_p^2 = 0.056$), significantly evident in PASC individuals that experienced severe acute disease ($p = 0.011$ and $p = 0.030$, to non-PASC with acute severe disease and PASC with acute mild disease, respectively; Fig. 5B). Furthermore, the analysis of the levels of IFN-β also revealed differences among groups (H = 7.84, $p = 0.020$, $\eta_H^2 = 0.09$). Indeed, we observed that PASC patients display higher levels of IFN-β than non-PASC group ($p = 0.037$; Fig. 5C). The IFN-β levels discriminates PASC that experienced severe infection with a F (1, 44) = 3.99, $p = 0.053$ and $\eta_p^2 = 0.095$, ($p = 0.013$ and $p = 0.042$ to non-PASC with acute mild disease and acute severe disease, respectively; Fig. 5D). Consequently, convalescent PASC patients presented a significantly higher IFN-β/IFN-α2 ratio when compared to non-PASC ($p = 0.0004$; Supplementary Figure 2). Overall, these results are consistent with a recent report showing higher levels of IFN-β in patients with PASC when compared to non-PASC group and healthy controls[60].

Finally, we observed a significant difference in IFN-λ levels in convalescent COVID-19 patients when compared to HC (H = 19.1,

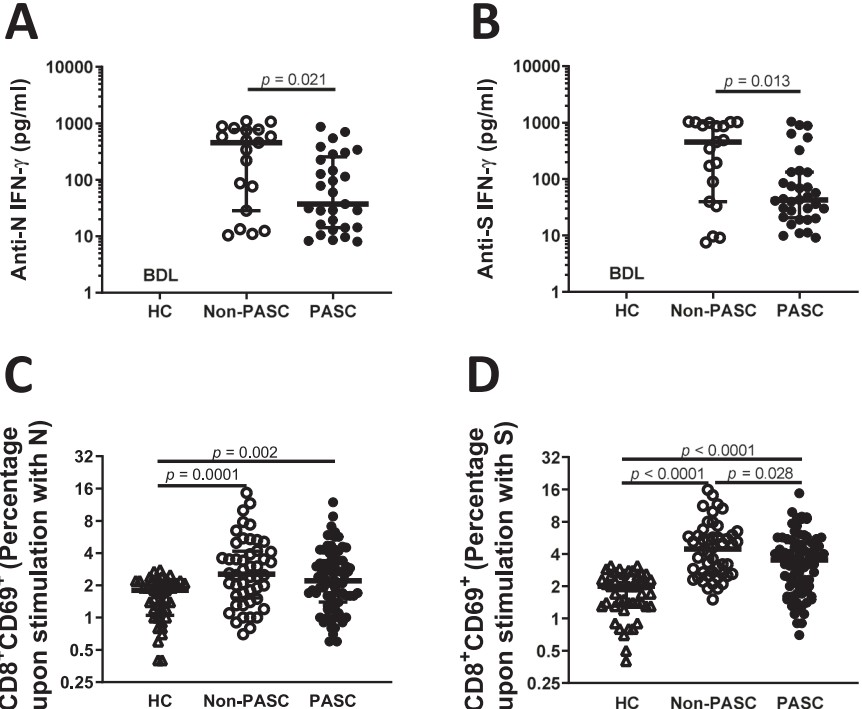

**Fig. 2 | Deficient memory response of CD8⁺ T cells of convalescent PASC patients.** Peripheral blood mononuclear cells from non-PASC (*n* = 65) and PASC (*n* = 62) individuals were stimulated with N or S viral peptides. Interferon-gamma was quantified by ELISA upon stimulation with N (**A**) and S (**B**) peptides for 24 h. All HC were tested, behaving below the detection limit of the technique. A Mann Whitney test was employed to compare the non-PASC and PASC groups. The surface expression of the early activation marker CD69 was evaluated upon 24 hours of stimulation with N (**C**) and S (**D**) peptides on CD8⁺ cells by flow cytometry. Data are shown in a scatter dot plot format as median ± IQR; Kruskal–Wallis test was applied to identify statistical differences followed by Dunn's multiple comparisons test.

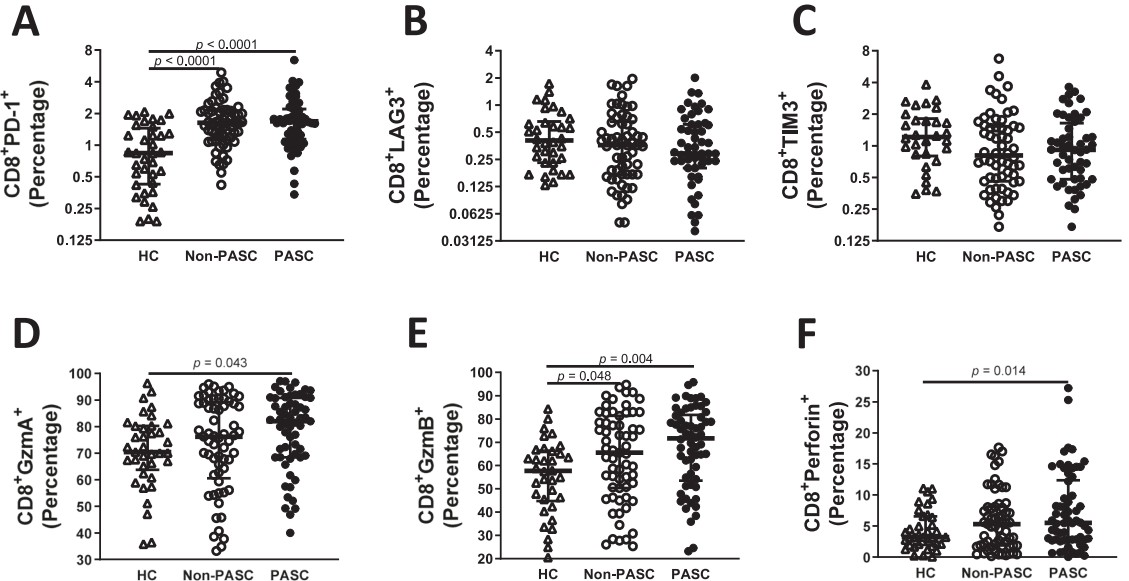

**Fig. 3 | Higher CD8⁺ T cell immune activation during SARS-CoV-2 infection on convalescents that developed Post COVID-19 condition.** (A) Percentages of the CD8⁺ T lymphocytes expressing PD-1 (**A**), LAG3 (**B**), TIM3 (**C**), granzyme A (**D**), granzyme B (**E**) and perforin (**F**) on PBMC of HC (*n* = 37), non-PASC (*n* = 65) and PASC (*n* = 62). Data are shown in a scatter dot plot format as median ± IQR; Kruskal–Wallis test was applied to identify statistical differences followed by Dunn's multiple comparisons test.

$p < 0.0001$, $\eta_H^2 = 0.13$). This increase is observed in patients developing or not symptoms ($p < 0.0001$ and $p = 0.002$, to Non-PASC and PASC, respectively; Fig. 5E). Interestingly, in the convalescent individuals, the IFN-λ 2/3 levels were positively correlated with the percentage of CD8⁺Eomes⁺ T cells in PASC patients (r(62) = 0.34; $p = 0.007$, Fig. 5F),

correlation not observed in non-PASC individuals. This could be consistent with viral persistence of SARS-CoV-2 that has been recently proposed, particularly in the intestine[80].

Considering this higher level of IFN-λ, which is central in the control of mucosal viral infection[81], we quantified the expression of the

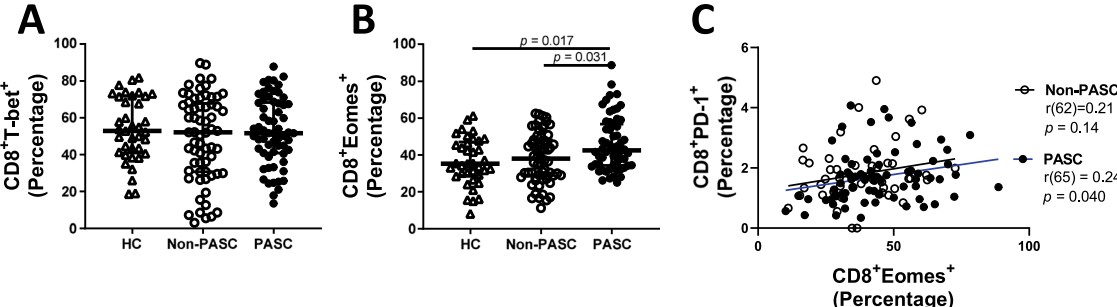

**Fig. 4 | Increased levels of EOMES in CD8⁺ T cells of convalescent PASC patients.** Percentages of Tbet⁺ (**A**), EOMES⁺ (**B**) CD8⁺ T lymphocytes on PBMC of HC ($n = 37$), non-PASC ($n = 65$) and PASC ($n = 62$) individuals. Kruskal–Wallis test was applied to identify statistical differences followed by Dunn's multiple comparisons test. Spearman's correlation between the percentage of CD8⁺PD-1⁺ and CD8⁺Eomes⁺ T lymphocytes for non-PASC and PASC individuals (**C**). Data are shown in a scatter dot plot format as median ± IQR.

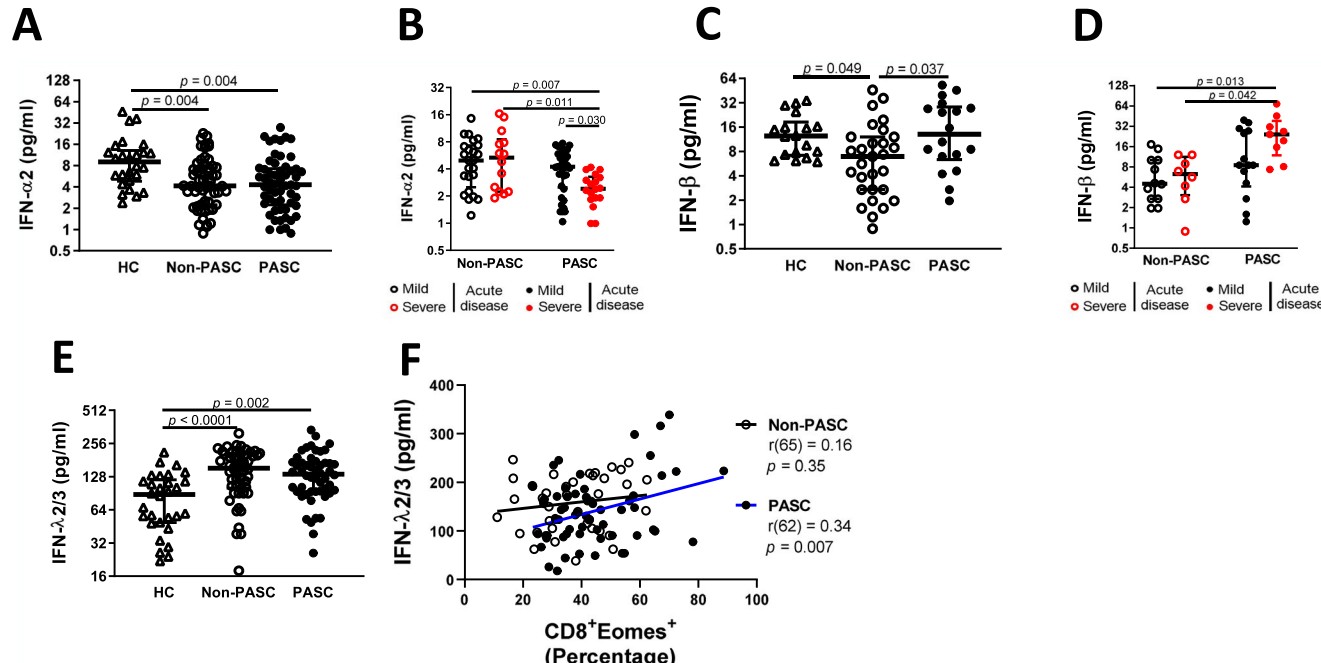

**Fig. 5 | Higher IFN-beta and IFN-lambda levels are associated with PASC.** The plasma levels of IFN-α2 (**A**), IFN-β (**C**) and IFN-λ2/3 (**E**) of HC ($n = 37$), non-PASC ($n = 65$) and PASC ($n = 62$) individuals and sub-divided by mild or severe acute disease (**B** and **D** for IFN-α2 and IFN-β, respectively). Spearman's correlation between the percentage of IFN-λ2/3 and CD8⁺EOMES⁺ for non-PASC and PASC individuals (**F**). Data are shown in a scatter dot plot format as median ± IQR. Kruskal–Wallis test was applied to figures **A**, **C** and **E** to identify statistical differences followed by Dunn's multiple comparisons test. A two-way ANOVA was used in figures **B** and **D**.

β7 integrin on CD8⁺ T cells, a useful surrogate in the blood for estimating intestinal T cell homing[82,83]. Of interest, statistical differences were found among groups (H = 8.21, $p = 0.016$, $\eta_H^2 = 0.047$). Our results indicated that both non-PASC and PASC individuals display lower percentages of CD8⁺β7 Integrin⁺ T cells when compared to HC ($p = 0.016$ and $p = 0.034$ for Non-PASC and PASC, respectively; Fig. 6A). Taking into consideration acute disease severity in this analysis, we found a significant interaction between PASC and acute disease severity (F(1, 127) = 5.26, $p = 0.024$, $\eta_p^2 = 0.041$). Interestingly, PASC with severe acute disease demonstrated a significant decrease of CD8⁺β7 Integrin⁺ T cells compared to those with mild severity ($p = 0.040$; Fig. 6B) suggesting the redistribution of the CD8⁺β7 Integrin⁺ T cells in the mucosa. Furthermore, our results indicated a strong negative correlation between Eomes and β7 integrin expression (Fig. 6C), which is valid to PASC and non-PASC patients (r(62) = −0.30 and $p = 0.019$ for PASC and r(65) = −0.32 and $p = 0.012$ for non-PASC; Fig. 6C). We also found a negative correlation between IFN-λ 2/3 levels

and β7 integrin expression for convalescent individuals (r(62) = −0.39 and $p = 0.001$ for PASC and r(65) = −0.32 and $p = 0.050$ for non-PASC; Fig. 6D). Altogether, our results suggest mucosal immune dysregulation in individuals with PASC.

### Specific IgA humoral response to SARS-CoV-2 antigens characterizes convalescent individuals presenting PASC

Considering the abovementioned profile of immune response including IFN-λ2,3, we hypothesized if viral antigens persist, the development of a specific IgA may occur, which are short-live immunoglobulins compared to IgG. Thus, we assessed the levels of the three types of immunoglobulins (IgA, IgM and IgG) against N and S viral proteins. As expected, after approximately six months of SARS-CoV-2 infection, convalescent individuals displayed high levels of IgG against N and S that contrast to the IgM response, which is rarely observed during chronic viral infections. No significant difference was observed for IgG response against N and S between PASC and non-PASC individuals

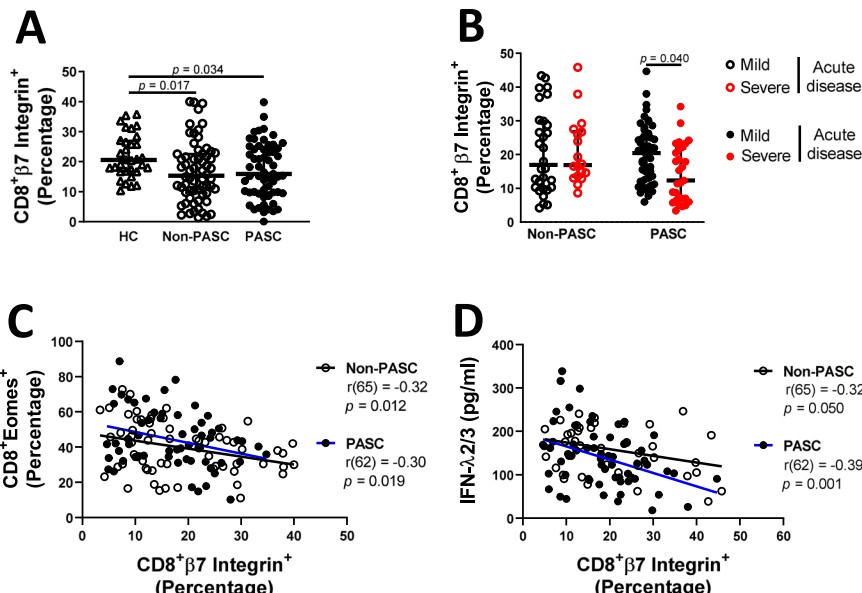

**Fig. 6 | Decreased levels of Integrin⁺ CD8⁺ T cells of convalescent individuals.** Percentages of β7integrin⁺ CD8 + T lymphocytes on PBMC of healthy controls ($n = 37$), non-PASC ($n = 65$) and PASC ($n = 62$) individuals (**A**) and sub-divided by mild or severe acute disease (**B**). Spearman's correlation between the percentage of CD8⁺β7integrin⁺ and CD8⁺ EOMES⁺ T lymphocytes for non-PASC and PASC individuals (**C**) and IFN-λ2/3 (**D**). Data are shown in a scatter dot plot format as median ± IQR; Kruskal–Wallis test followed by Dunn's multiple comparisons test and a two-way ANOVA were applied, respectively, to figures **A** and **B**.

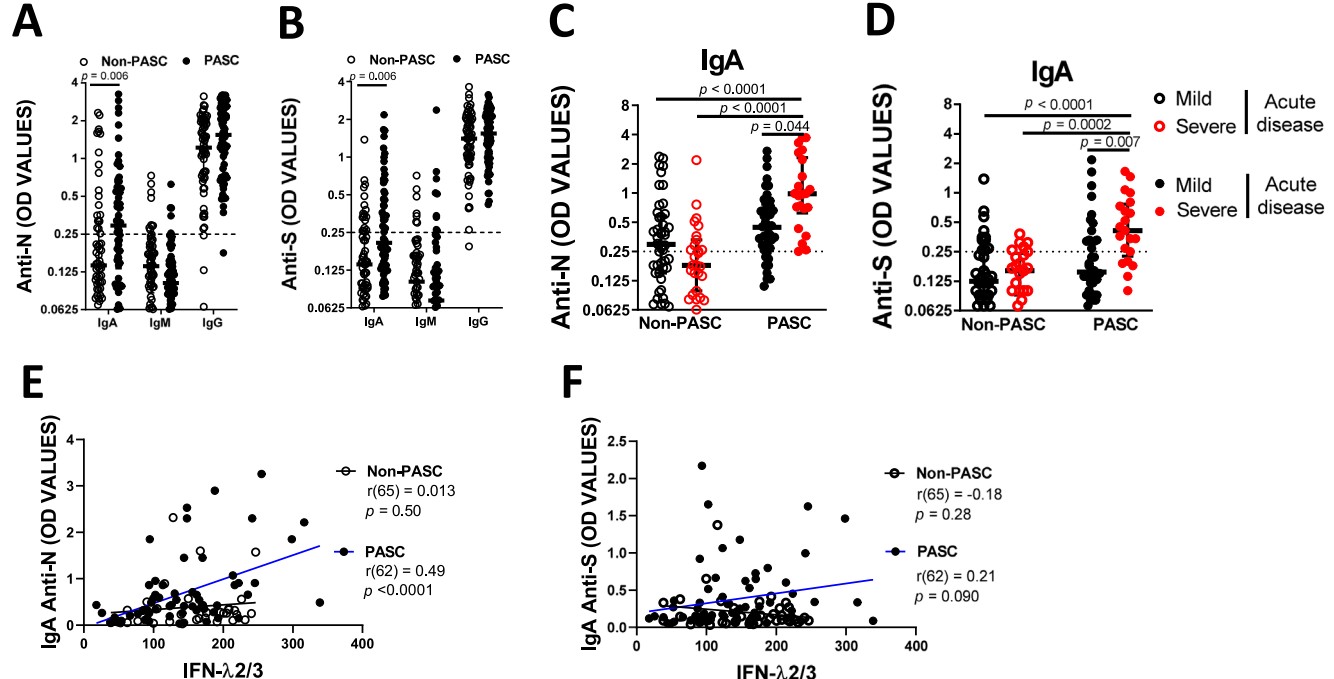

**Fig. 7 | Type III interferon and IgA signatures on Post COVID-19 condition.** Quantification of the anti-N (**A**) and anti-S (**B**) IgA, IgM and IgG response after SARS-CoV-2 infection divided by individuals that developed ($n = 62$) or not PASC ($n = 65$) six months after acute disease. Quantification of anti-N (**C**) and anti-S IgA (**D**) at six months after SARS-CoV-2 infection on non-PASC ($n = 65$) and PASC ($n = 62$) individuals divided by the severity of acute disease. Spearman's correlation between the percentage of Anti-N IgA (**E**) or anti-S IgA (**F**) and IFN-λ2/3 for non-PASC and PASC individuals. Data are shown in a scatter dot plot format as median ± IQR; Mann Whitney test and a two-way ANOVA were applied, respectively, to figures **A**, **B** and **C, D**.

(Fig. 7A, B). Interestingly, our results demonstrated that more than 56.5% and 46.8% of PASC individuals have specific IgA to N and S, when compared to 30.8% and 26.2% of non-PASC individuals. Thus, IgA levels were significantly higher in PASC patients when compared to non-PASC group (U = 1308, $p = 0.006$, r = 0.30 for anti-N and U = 1446, $p = 0.006$, r = 0.24 for anti-S, Fig. 7A, B). As before, we assessed if the severity of acute disease presented interaction with PASC occurrence performing

a two-way ANOVA: F(1,126) = 4.52, $p = 0.036$ and $\eta_p^2 = 0.036$ for IgA anti-N and F(1,126) = 3.70, $p = 0.047$ and $\eta_p^2 = 0.029$ for IgA anti-S. Indeed, the observed increase on anti-viral IgA on convalescent PASC patients was specific to those who previously developed severe disease, against either the N protein ($p < 0.0001$, $p < 0.0001$ and $p = 0.044$ to non-PASC with mild disease, non-PASC with severe disease and PASC with mild disease, respectively; Fig. 7C) or the S protein ($p < 0.0001$, $p = 0.0002$

and $p = 0.007$ to non-PASC with mild disease, non-PASC with severe disease and PASC with mild disease, respectively; Fig. 7D). This interaction between the presence of PASC and acute disease severity was not observed in IgM ($F_{(1,126)} = 0.49$, $p = 0.487$ and $\eta_p^2 = 0.004$ for anti-N and $F_{(1,126)} = 0.051$, $p = 0.821$ and $\eta_p^2 < 0.001$ for anti-S, Supplementary Figure 3A-B) or IgG levels ($F_{(1,126)} = 0.30$, $p = 0.59$ and $\eta_p^2 = 0.002$ for anti-N and $F_{(1,126)} = 0.43$, $p = 0.51$ and $\eta_p^2 = 0.004$ for anti-S, Supplementary Figure 3C-D). By plotting IFN-λ 2/3 levels with the levels of IgA anti-N, we found a strong significant correlation in PASC patients, but not with non-PASC patients ($r_{(62)} = 0.49$ and $p < 0.0001$ for PASC and $r_{(65)} = 0.013$ and $p = 0.50$ for non-PASC; Fig. 7E). A similar tendency was observed with the levels of IgA anti-S and IFN-λ 2/3 although not significant ($r_{(62)} = 0.21$ and $p = 0.090$ for PASC and $r_{(65)} = -0.18$ and $p = 0.28$ for non-PASC; Fig. 7F).

Overall, our results demonstrated that patients with PASC are characterized by a specific IgA humoral response against SARS-CoV-2 antigens.

### Patients with post-COVID-19 condition displayed an exacerbated inflammatory signature during acute infection

Having observed persistent immune reaction in PASC individuals and considering that most of these differences revealed an interaction with acute disease severity, we then addressed whether the acute phase of infection may have an impact on PASC. The levels of circulating IL-6, IL-8/CXCL8 and IP-10/CXCL10 were analysed in plasma samples collected during acute SARS-CoV-2 disease and compared to HC. Indeed, previous studies have defined IL-6, IL-8/CXCL8 and, IP-10/CXCL10 as early prognostic parameters of COVID-19 severe disease[84–86]. Significant differences were found between groups for IL-6 ($H = 16.03$, $p < 0.001$, $\eta_H^2 = 0.21$), IL-8 ($H = 26.0$, $p < 0.0001$, $\eta_H^2 = 0.26$) and IP-10 ($H = 52.61$, $p < 0.0001$, $\eta_H^2 = 0.53$) levels (Fig. 8A–C). Indeed, our results highlighted that this trio of cytokines is significantly higher in convalescent patients that developed PASC at six months post-infection when compared to non-PASC or HC ($p = 0.043$ and $p < 0.0001$ for IL-6 and $p = 0.049$ and $p < 0.0001$ for IL-8 compared to non-PASC and HC, respectively; Fig. 8A, B). The plasma levels of IP-10 were significantly increased on both non-PASC and PASC when compared to HC ($p < 0.0001$ for both; Fig. 8C). Overall, our data demonstrates increased levels of inflammatory mediators on the plasma of acute SARS-CoV-2 patients that later developed PASC. All together, these results suggest that patients with acute severe disease, in which inflammation is greater, can be associated with higher risk to develop PASC.

### Discussion

Herein, we have addressed the dynamics and immune activation of CD4+ and CD8+ T cells in convalescent individuals. We observed that CD8+ T cells display higher levels of immune dysregulation compared to CD4+ T cells. Of interest, we found that several immune aspects are not only occurring in individuals having PASC form but also in non-PASC individuals six months after infection, which supports that immune alterations due to SARS-CoV-2 infection persist longer as it was initially expected. The most important information relates to the fact that CD8+ T cells remain activated after six months presenting higher levels of perforin and Eomes expression in PASC individuals. Secondly, we found an attrition of the CD8+ β7 Integrin+ T cell population in peripheral blood of PASC individuals, concomitantly with higher plasmatic levels of IFN-λ2/3, and the presence of specific IgA to SARS-CoV-2 antigens. Finally, by analysing retrospectively the plasma of these individuals at the early phase of infection, we showed that the extent of inflammation is associated with the occurrence of later PASC symptoms.

Thus, we looked at possible biological signatures of post-acute sequelae SARS-CoV-2 condition that may reflect physiopathology and help on the diagnosis and treatment of this syndrome as well as at mechanisms that may connect acute disease and PASC. This cohort is composed of 127 patients, of which 81% had been hospitalized during their acute disease. In accordance with others[15,18,27,37,87–89], we did not find any influence of sex or any specific comorbidity in the development of PASC. When looking at both CD4+ and CD8+ T cell populations, at six months after infection, we noticed that convalescent patients did not display lymphopenia, with full recovery of CD4+ T cells. This contrasts to the acute phase in which lymphopenia is a predictive marker of disease severity and the occurrence of CD4+ T cell apoptosis is a confounding marker of the pathogenicity[90,91]. Timing of observation seems crucial, as recent works described accordingly that the alterations observed in CD4+ T cells (and B cells) last for only a few weeks after acute disease and disappear in a short amount of time[87,92–94]. In opposition to CD4+ T cells, our results highlighted that CD8+ T cells remain altered for a longer period, irrespective of their symptoms during acute disease. Our results are consistent with previous reports showing that months after COVID-19, CD8+ T cells are activated and expressing a cytotoxic profile, with increased expression of perforin and granzymes, which may be related to patients' long-term outcome[60,92]. Furthermore, other groups have also reported a significant increase in exhausted markers such as PD-1 in CD8+ T cells[60,93,95]. Therefore, this may appear contradictory to the notion that exhausted T cells expressing PD-1 may express lower levels of effector molecules such as perforin and granzyme B as well IFN-γ[96]. However, PD-1 is expressed after T cell activation and shown to promote, in certain condition, effector memory CD8+ T cells[64–66]. Furthermore, our results indicate higher levels of perforin as well of Eomes in PASC individuals, which is more consistent with an activated phenotype instead of an exhausted one. As supportive evidence, we found that Eomes is positively correlated with PD-1 in PASC individuals.

Our results also demonstrated that COVID-19 patients, particularly those with PASC, display (i) higher levels of IFN-β and IFN-λ (this latter being associated with mucosal microbial defences) and

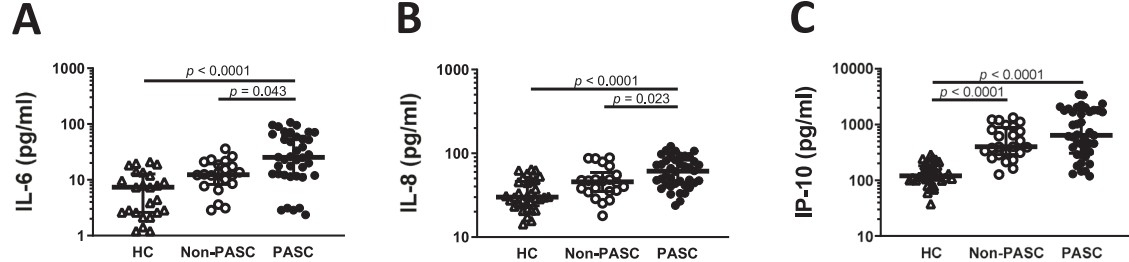

**Fig. 8 | Post-COVID-19 condition associates with a higher inflammatory signature during acute disease.** The levels of IL-6 (**A**), IL-8 (**B**) and IP-10 (**C**) were quantified on the plasma of HC ($n = 37$) and SARS-CoV2 infected patients that develop (PASC, $n = 36$) or do not develop (Non-PASC, $n = 37$) post COVID-19 condition. Data are shown in a scatter dot plot format as median ± IQR; Kruskal−Wallis test was applied to identify statistical differences followed by Dunn's multiple comparisons test.

(ii) specific anti-S IgA, which are short-lived Ig compared to IgG, and mostly induced in mucosal tissues. Altogether, these observations associated with lower levels of blood CD8[+]β7 Integrin[+] T cells, a marker of circulating mucosal cells strongly suggest the persistence of SARS-CoV-2 in mucosal tissues. Recent reports based on tissue biopsies or autopsies[97,98] have observed the presence of SARS-CoV-2 genomic material in human tissues for at least four months after acute infection, particularly in the gastrointestinal system, the nervous system and other ACE2-rich tissues[49–54]. Thus, regarding CD8[+] T cells, this continuous viral replication induces CD8[+] T cell activation associated with local and/or systemic manifestations[52,53,57]. Thus, IFN-λ levels correlated positively with the percentage of CD8[+]Eomes[+] T cells and inversely with those expressing β7 integrin, particularly in PASC individuals. IFN-λ is mainly produced by the epithelial tissue of intestine and lungs[99,100]. We observed that PASC patients had a lower level of IFN-α2 but a higher level of IFN-β. Type I interferons are crucial in eliciting an effective antiviral response, and therefore our results may emphasise different sources of innate cells, whereas they may induce different interferon-stimulated genes in COVID-19 individuals[101]. Thus, the absence of IFN-α2 in PASC could contribute to the absence of viral control. It has been proposed that low levels of type I interferons during the acute disease are associated with increased viral load and disease severity[101,102]. Other reports have suggested that escapes to IFN-α response is associated with the presence of autoantibodies against IFN-α or SARS-CoV-2 antagonist proteins[102–104]. Our data highlight the relevance of type I and II Interferons during disease reinforcing previous publications that have shown elevated expression of the cytokines during acute disease and in convalescent patients presenting PASC[60,105,106]. By a retrospective analysis, we found that patients developing PASC are those who displayed greater inflammation (IL-6, IL-8 and IP-10) during acute disease. However, the levels of these cytokines and other biological parameters such as Ferritin, D-Dimer and CRP returned to normality during convalescence, independently of the development or not of PASC (Supplementary Figure 4). A repeated measures analysis of all variables comparing acute with convalescence samples did not show any statistical significance (Supplementary Table 2). Of interest, we recently reported that the levels of IP-10 (CXCL10) correlated positively with disease severity and CD4 T cell lymphopenia related to higher levels of T cell apoptosis[90]. These T cell defects may contribute to the absence of viral control by immune cells favouring viral dissemination and therefore leading to viral persistence several months after infection. It is important to mention that beside SARS-CoV-2-specific CD8 T cell, our data strongly suggest a T-cell receptor-independent activation of bystander CD8[+] T cells. Although without specificity for the virus, the bystander CD8[+] T cells have been demonstrated in other viral infections to impact the course of the immune response[107], including being responsible for causing collateral damage to the host[108]. Thus, determining whether such early immune T cell defect is associated with PASC may be important to treat patients early after hospitalization limiting side effects.

Consistent with viral persistence in the mucosa, we detected in PASC patients higher levels of IgA directed against the S and N proteins, when compared to the other group, a difference that was more prominent in the patients who had the severe acute disease. To our knowledge, this is the first description of such a specific humoral IgA response characterizing PASC. A previous study showed that patients with lower titres of SARS-CoV-2 specific antibodies during early recovery are more likely to develop post-acute sequelae SARS-CoV-2 condition[109]. Other groups have reported that patients with persistent symptoms have a lower specific humoral response[110–112]. Herein, we found a strong significant positive correlation between IFN-λ and the levels of IgA in PASC individuals. No difference was observed for the IgG response. In contrary to total IgGs, which have a half-life of 26 days, IgA in blood circulation have a half-life of six days[113–115]. Thus, the observation of high levels of viral-specific IgA at six months post-acute

infection supports the hypothesis of continuous viral replication, which will stimulate and maintain plasmablast cells. Thus, monitoring IgA may represent an easy way of monitoring chronic COVID-19-infected patients.

Our results have clearly established critical immune parameters associated with patients with PASC compared to non-PASC. The percentage of symptomatic patients in our cohort (49%) is in accordance with a recent systematic review (30 to 50% of post-COVID condition six to twelve months after acute infection), especially considering the six months median follow-up and high percentage of patients that required hospitalization[112]. Also, we are perfectly aligned in terms of symptoms reported as most of the studies and meta-analysis point out fatigue as the most common complaint, usually followed by dyspnoea and neurological or psychiatric symptoms[11,17–19,27,37,40,116–119]. Our results also highlighted that even in patients without major symptoms, six months after infection, the immune status of individuals not fully recapitulates to that observed in healthy donors suggesting that it is of crucial importance to monitor in a more accurate manner immune parameters associated with SARS-CoV-2 infection.

We recognize several limitations of our study. Most importantly, we are studying a condition whose diagnosis is based on symptoms report and clinical judgement. Therefore, despite all the efforts to accurate classifications, we shall admit a certain level of subjectivity in the process. Secondly, our study is exploratory, and despite pointing to viral persistence in the mucosa as a very important mechanism of disease, it was not designed to unequivocally demonstrate it. Further studies may look deeper into the early phase of the disease, including more extensive analyses of anti-SARS-CoV-2 humoral and adaptive T-cell specific responses, and its consequence on long term. Thus, a recent study suggested low perforin expression in CD8[+] T cells during the acute phase is associated the persistence of symptoms[120]. Also, once our cohort is mainly constituted by hospitalised patients, it would be ideal to have enlarged our sample to include more patients who never required hospitalisation. However, for all the statistical tests applied, the effect size was large according to Cohen's classification supporting the robustness of our data irrespectively to sample size.

In conclusion, our results display major advances in our understanding of PASC in which parameters of immune activation (CD8[+] β7 Integrin[+] T cells and IgA) are consistent with viral persistence and able to characterise these patients (Supplementary Figure 5). If viral persistence is inferred by other groups as the cornerstone of PASC, treatment of this syndrome may be based on antiviral drugs, which have not yet been tested.

## Methods
### Patients and study design
This is a single-center prospective cohort study, performed at Hospital de Braga (HB), a tertiary Portuguese Hospital. From 21st September 2020 to 26th February 2021, patients admitted due to COVID-19, confirmed through a PCR positive nasopharyngeal swab, were invited to participate in the study. In respect to the Portuguese Law 21/2014, our research complies with the local Ethics Committee of Braga Hospital that approved the study with the reference 123/2020 (approved on 09/09/2020). Volunteering participants gave written informed consent in compliance with the Declaration of Helsinki principles. Sex or gender was not considered during study design or recruitment. According to clinical protocol, blood samples were collected at admission and on each 72 hours, throughout hospitalisation, until discharge or suspension of oxygen therapy. Only the blood sample corresponding to the worst disease point (defined as the highest respiratory support during hospitalisation) was used for cytokine quantification and as a reference of patients' acute disease. Patients who refused to participate or were not able to give their informed consent, who had been previously vaccinated against COVID-19 or had previous autoimmune disease, immunodepression, active solid cancer (stage III or IV) or

haematological malignancy, treated with chemotherapy or immuno-suppressants higher than prednisolone 20 mg or its equivalent were not included in the study. Also, patients with evidence of any simultaneous bacterial infection since admission or during their hospitalisation were excluded from that timepoint. After discharge, the abovementioned patients had a follow-up consultation six months after acute disease (Median 165 days). Other convalescent individuals, referred to Post COVID-19 consultation, who were not hospitalised during acute disease or were not recruited during their hospitalisation, were recruited in the outpatient setting (from September 2020 to June 2021), and had their blood collected in a similar interval of time between acute disease and consultation. Of those patients, only clinical data, and basic laboratory results (which had to include a PCR-positive nasopharyngeal test) of their acute disease were available. Inclusion and exclusion criteria were maintained.

It was anticipated that patients reporting on the consultation acute symptoms compatible with any ongoing infection (COVID-19 reinfection or others) would be excluded from the study. No COVID-19 test was performed on the consultation due to the absence of acute symptoms.

A control group was constituted of patients who had their blood collected before elective diagnostic exams to which they needed a PCR-negative SARS-CoV-2 test in the last 48 hours. Only non-vaccinated, nor previously infected, nor symptomatic patients were eligible, and the exclusion criteria described above were applied. Sera of those patients were analyzed to exclude the presence of antibodies related to previous unnoticed infection. Only two out of thirty-nine healthy controls were excluded from this study considering their positivity for the presence of anti-S and anti-N IgG.

### Data collection
For each patient, data were collected from the medical records and inserted into our study database. Variables comprised demographics, major comorbidities, dates of onset, diagnosis, hospital admission and discharge. At admission and during hospitalization, daily information on the disease stage, need of respiratory support, treatments used, diagnosis of pulmonary embolism if present, tomography data, blood count, quantification of C-reactive protein (CRP), ferritin, d-dimer, and procalcitonin, and clinical observation of patients' evolution were collected. At the outpatient setting, and according to clinical protocol, patients were asked about the existence or persistence of symptoms attributable to PASC and an extensive physical examination was performed. Additional exams directed to patients' complaints were requested when necessary. Patients that could benefit the most were also referred to rehabilitation programs. Patients were grouped according to the presence or absence of PASC diagnosis, established in accordance with the WHO published criteria[25]. Our clinical data complies to the STROBE guidelines.

### PBMC isolation and stimulation
Blood samples were collected using Heparin Blood Collection tubes (VACUETTE). After transportation to ICVS, samples were processed in BSL2 laboratories. PBMC were isolated using equal volumes of peripheral blood and Histopaque 1077 (MilliporeSigma, St Louis, Missouri, USA). After centrifugation, plasma samples were collected and stored (−80 °C). PBMC were frozen in Fetal Bovine Serum (FBS, Gibco, Thermo Fisher Scientific) with 10% of DMSO. Samples were defrosted and centrifuged at 250 g for 5 minutes to remove DMSO. Upon suspension in complete RPMI medium (RPMI-1640 culture medium supplemented with 2 mM glutamine, 10% FBS, 10 U/mL penicillin/streptomycin and 10 mM HEPES (Gibco, Thermo Fisher Scientific), PBMC were seeded at a concentration of $1 \times 10^6$ cells / mL in a 96-well plate (Corning, NY) and incubated for 24 h with 500 mg/mL of purified anti-human CD28 antibody (ref. 302902, Biolegend, CA, USA) and 500 mg/mL of purified anti-human CD40 antibody (ref. 334302,

Biolegend, CA, USA) along with 1 µg/mL of SARS-CoV-2 Spike Glycoprotein (ref. RP30020, Gene Script, USA) or SARS-CoV-2 Nucleoprotein (ref. RP30013, Gene Script, USA). After incubation, supernatant was stored for cytokine quantification and cells were characterised by flow cytometry.

### Phenotypic analysis of peripheral blood mononuclear cells
PBMC were thawed and divided in two 96-well plates for immune phenotyping. Surface staining was performed with the following antibodies: CD3 (clone SP34-2), from BD Biosciences, CD4 (clone OKT-4) and β7 integrin (clone FIB504) from Invitrogen, CD8 (clone SK1) and PD-1 (clone EH12.2H7) from Biolegend and LAG3 (clone P18627), and TIM3 (Clone 344823) from R&D. Intracellular staining was performed using the Foxp3 / Transcription Factor Staining Buffer Set (Ref. LTI 00-5523-00, Invitrogen) according to manufacturer's instructions and using the following antibodies: granzyme A (clone CB9), T-bet (clone eBio4B10) and Eomes (clone WD1928) from Invitrogen and Granzyme B (clone GB-11) from Sanquin and perforin (clone Pf-344) from Mab-Tech. Cells from the stimulation assay were characterised through surface staining using the following antibodies: CD3 (clone SP34-2), from BD Biosciences, CD4 (clone OKT-4) from Invitrogen, CD8 (clone SK1) and CD69 (clone FN50) from Biolegend (Supplementary Table 3). Samples were acquired on LSRII flow cytometer (BD Biosciences) using the DIVA Software and data was analysed using FlowJo software. Gating strategies are shown in Supplementary Figures 6–8.

### Cytokine quantification
Cytokine quantification of plasma samples was performed with LEGENDplex™ Human Anti-Virus Response Panel (13-plex) (ref. 740390, Biolegend, CA, USA), according to manufacturer's instructions. IFN-γ produced upon peptide stimulation was quantified in cell culture supernatant using an ELISA MAX Human IFN-γ (ref. 430104, BioLegend, CA, USA), according to the manufacturer's instructions.

### Production of the nucleoprotein of SARS-CoV-2
The full gene coding for the N protein (accession number YP_009724397, amino acids 1 to 419) was cloned into a pET24d vector (Eurogentec). The protein was expressed in *E. coli* bacteria were transformed with this plasmid and the N protein production was induced by the addition of 0.5 mM IPTG. After four hours at 37 °C with shaking at 130 rpm, the cells were lysed (buffer A, 50 mM Tris, 500 mM NaCl and 5% glycerol, pH = 7.5) and sonicated for 4 minutes. The supernatant was passed through a 1 mL nickel-NTA resin (GE, 17-5318-02). The protein was diluted in buffer B (50 mM Hepes, 150 mM NaCl, 10 % glycerol, pH = 7) and further purified on a 1 mL Source 15 S (GE, 17-0944-10) cationic exchange chromatography. The protein N was eluted by applying a gradient with buffer C (50 mM Hepes, 1 M NaCl, 10 % glycerol, pH = 7). The sample was then concentrated to 2 mL and applied onto a preparative gel filtration column (GE, 17-1069-01) equilibrated in buffer D (50 mM Tris, 200 mM NaCl, 10% glycerol, pH = 7.5). The N protein was eluted as an oligomer.

### IgM, IgA, and IgG humoral responses
The antibody production was monitored by measuring specific Igs by enzyme-linked immunosorbent assay (ELISA) against the N and S1 proteins (Sars-Cov-2 S1 protein carrier free BioLegend). NUNC Max-iSorp™ well plates were coated overnight with antigens (0.5 µg/ml in Tris/Hcl pH 9.6). After washing and saturation with fetal bovine serum (BSA), plasma was serially diluted and incubated for 120 min. Plates were then washed and incubated with goat anti-Human IgG (Fc specific)-peroxidase (A0170, Millipore Sigma), goat anti-Human IgM (Fc specific)-peroxidase (401905, Millipore Sigma), and goat anti-Human IgA (Fc specific)-peroxidase (SAB3701229, Millipore Sigma) for 120 min. Secondary antibodies were titrated to optimize sensitivity. Following several washings, substrate reagent solution (R&D systems)

was added for 30 min. The reactions were stopped using sulfuric acid (0.4 N). The plates were read on a Thermo Scientific™ Varioskan™ reader at wavelengths of 450 nm and 540 nm. Thus, after serial dilutions of the plasma, results shown in the figures are 1/800 for IgM and 1/400 for IgA and IgG.

## Statistical analysis

Statistical analysis were performed using SPSS version 28 software (IBM, New York, USA) and data was plotted using GraphPad Prism version 8 software (San Diego, CA) For each variable, a two-way ANOVA was used to assess the interaction between disease severity during acute disease and the occurrence or absence of PASC. Only variables where the interaction terms were significant were divided according to these factors; if these significant differences were not observed, only the factor occurrence or absence of PASC was evaluated. Regarding the small sample size and the non-normality observed in our variables, Kruskal–Wallis test was applied to identify statistical differences. For variables that attained global significance, pairwise comparisons were performed using Dunn's multiple comparisons test. A Mann-Whitney test was employed when the comparison of Non-PASC and PASC groups was performed. The chi-square test was performed for categorical variables to assess the dependence between variables. For each statistical test we calculated the adequate effect size measure (eta squared for two-way ANOVA ($\eta_p^2$) and Kruskal-Wallis test ($\eta_H^2$), r for the Mann-Whitney test and the V-Cramer / Phi for the Chi-square test) and its interpretation followed the Cohen's effect size cut-off values[121]. Correlations were measured using Spearman's correlation coefficient.

## Reporting summary

Further information on research design is available in the Nature Portfolio Reporting Summary linked to this article.

# Data availability

The data generated in this study are provided in the Source Data file. Patients-related data were generated as part of clinical examination and may be subject to donor confidentiality. Source data are provided with this paper.

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

## Acknowledgements

This work has been funded by National funds, through the Foundation for Science and Technology (FCT) - project UIDB/50026/2020 and UIDP/50026/2020 and by the project NORTE-01-0145-FEDER-000039, supported by Norte Portugal Regional Operational Programme (NORTE 2020), under the PORTUGAL 2020 Partnership Agreement, through the European Regional Development Fund (ERDF) to R.S., the FCT contracts 2021.07836.BD to AMF and CEECIND/00185/2020 to RS, and the Clinical Academic Center (2CA-Braga) grant to A.S.C. Marne Azarias thanks the ANRS | Maladies infectieuses emergentes for his fellowship. This work was also supported by a grant to J.E. from the Fondation Recherche Médicale and the Agence Nationale de la Recherche (COVID-I²A). J.E. is supported by Canadian Institute of Health Research (FRN-177760) and the Canada Research Chair program.

## Author contributions

A.S.C. designed the study, collected biological samples and clinical data, formalized the patient agreement, communicated with the hospital, acquired funding, and wrote the manuscript. A.M.F. coordinated the biological specimen processing and storage and performed the T cell phenotyping flow cytometry experiments, cytokine quantification, data analysis and wrote the manuscript. M.A-da-S. and S.A. performed the quantification of the anti-SARS-CoV-2 humoral response. P.C. overview the statistical analysis. A.I.O, O.P., M.M., B.O., M.B., J.R.L., R.D., R.C., L.N.S., A.R.M., C.A., A.C. and C.C. collected biological samples and clinical data, formalized the patient agreement, and communicated with the hospital. J.P. and A.G.C. participated in study design and logistics; J.E. provided coordination, supervision and acquired funding. R.S. designed the study, analyse the data, acquired funding, and wrote the manuscript.

## Competing interests

The authors declare no competing interests.

## Additional information

[1]Life and Health Sciences Research Institute (ICVS), School of Medicine, University of Minho, Braga, Portugal. [2]ICVS/3B's – PT Government Associate Laboratory, Braga/Guimarães, Portugal. [3]Department of Internal Medicine, Hospital of Braga, Braga, Portugal. [4]Clinical Academic Center-Braga, Braga, Portugal. [5]INSERM-U1124, Université Paris Cité, Paris, France. [6]CHU de Québec - Université Laval Research Center, Québec City, Québec, Canada. [7]These authors contributed equally: André Santa Cruz, Ana Mendes-Frias. ✉e-mail: andrescjoao@med.uminho.pt; estaquier@yahoo.fr; ricardosilvestre@med.uminho.pt

