## [Peer Review File · Nature Communications]

Post-acute Sequelae of COVID-19 is Characterized by diminished peripheral CD8⁺β7 Integrin⁺ T cells and Anti-SARS-CoV-2 IgA responseREVIEWER COMMENTS

Reviewer #1 (Remarks to the Author):

This is a very nice paper assessing the immune response in convalescent individuals with PCC compared to convalescent asymptomatic and uninfected individuals. Some comments on statistical aspects of the paper:

Major:

1. The authors mention on line 264 that the healthy controls were age- and gender-matched with the PCC and non-PCC cases. A description of the matching needs to be in the Patients and Study Design section of the paper.
2. The statistical methods used should take into account the matched design of the study (they currently do not).
3. There are numerous statistical tests performed throughout the paper for the different outcome measures. Multiple testing corrections need to be used to control for the type-1 error rate.

Minor:

Table 1: length of stay in hospital - please specify the unit of measure? I assume it is days? Also, length is misspelled.

Reviewer #2 (Remarks to the Author):

There is a need to understand physiological parameters and molecular signatures associated with long COVID. This manuscript highlights findings from an immunological profile analysis of adults with and without post-acute sequelae of COVID-19, referred to herein as post COVID condition (PCC). The peripheral blood mononuclear cells were examined for responsiveness to N and S antigens, as well as mucosal IgA levels, and plasma soluble factors/cytokines following SARS-CoV-2 infection from adults in both acute and convalescent stages. The overall goal was to explore immune cell marker indicators associated with development of post-COVID-19 conditions, such as fatigue and dyspnea. The exploratory nature of the study was clearly stated, but it seems some parameters and analysis was not fully disclosed. There was limited description regarding how PCC was established clinically across adults enrolled, and there did not seem to be any repeated measures statistical analysis completed for the two timepoints sampled (for any of the endpoint markers). There was additional lack of clarity on whether or not there were negative results, or lack of associations with immune markers, which is still important contributions to the emerging field of study. The findings for the persistent activated CD8 T cell subsets and higher levels of serum IgA was an interesting finding, but somewhat confusing to follow when discussed as compared to healthy controls. A major concern was that the healthy adult cohort demographics were not provided in Table 1, which makes this comparison a questionable control group. There were also some grammatical and spelling errors that merit attention during revision. A list of major and minor concerns are provided below to improve the overall clarity of the approach, results and conclusions.

Major concerns

1. Line 105: "In this study, we explored the immune response associated with the persistence of symptoms upon acute SARS-CoV-2 infection", re-write in Intro as a main objective, and provide a stated hypothesis? Were there other immunological measures explored that were omitted from these results? Understanding the immune parameters with negative/lack of association are also important to consider alongside the signature that is positive.
2. Was the process for diagnosis/documentation of PCC collected in the same manner for each individual included in this cohort investigation? It seems that some PCC individuals could fall under two category types of PCC, thus have both fatigue and dyspnea. Please clarify this in cohort

description.

3. For the PCC in this cohort, were correlations measured between the IgA or CD8 T cell finding with these subsets of PC symptoms/conditions. This is relevant to understand the specificity of this immunological alteration to the myriad types of PCC reported in the literature. What falls under the category of "other" for the PCC.

4. Line 248-249. No demographics provided in Table 1 for the healthy controls (HC). Overweight should be added, with BMI cut off stated for obesity. Without these data on HC, the comparisons in the figures are difficult to understand for being matched or controlled for important variables such as age, sex, BMI, etc. The lack of HC demographics reduces enthusiasm for the rigor of the Data described across fig 4, 5, 6, 8

5. The Title does not fully depict the changes being measured, and title merits inclusion of the country cohort (Portuguese adults). Further, the term "altered" is unclear relative to the direction (increased?) markers of activation.

a. Perhaps reword the finding to the title "Blood CD8+beta7 T cells and serum IgA humoral response against SARS-CoV-2 antigens characterized post COVID-19 condition at 6 months post infection in adults"

6. Were the patients/participants screened for SARS-CoV-2 positivity at enrollment and follow up visit (6 months)? The potential for re-infection was possible during these time periods and could be a factor with respect to the results reported at convalescent, and this should be clarified, if not verified. Please add the documentation for PCR/antigen positive test results to this to methods description for the acute phase as well as convalescent visits.

7. Did the stimulation of PBMCs from acute phase blood collection occur for N and S responsiveness, and test for differences between PCC and non-PCC? This would be a repeated measures comparison of acute and convalescent responses instead of only in convalescent stages. Given the large number of hospitalized patients included, these data are likely biased by many having moderate/severe acute disease conditions.

Figure 1 and 2: the statistical comparisons in PBMC responses between PCC and Non-PCC, versus those from healthy controls are somewhat confusing in the results text and misleading. How was lymphopenia in the acute phase taken into consideration for influencing results?

a. Line 256-257 states "We did not observe any difference in PCC prevalence according to acute disease severity, need of hospitalization, length of stay, lymphocyte count or commonly used inflammation markers."and unclear if this includes the PBMC stimulation response in acute stage?

8. The finding that patients show distinct profiles in type I IFN, and mucosal type III compared to uninfected individuals supports existing publications. Please bolster support for this and reduce emphasis on this as a novel finding.

<https://www.science.org/doi/10.1126/science.abc6027>

<https://www.nature.com/articles/s41467-021-27318-0>

Given this, was a sample size calculation completed? This would add statistical rigor to the overall results.

9. In methods "At admission and during hospitalization, daily information on the disease stage, need of respiratory support, treatments used, diagnosis of pulmonary embolism if present, tomography data, blood count, quantification of C-reactive protein (CRP), ferritin, d-dimer, and procalcitonin." they are missing from the 6 month follow up analysis.

a. What sort of follow up PCC vs Non-PCC analysis of these key clinical parameters was performed? Notably, these analytes also merit further explanation in the post COVID condition for

associations to the recovery phase, alongside immunological findings (IgA, CD8 signature etc.)?

10. Eligibility criteria: were all patients required to provide a PCR or antigen test positive report, and if yes, these would have dates to indicate time frame of symptoms/active infection, but that information is limited in this report. Is acute phase within days 1-15 from test positive result or symptoms?

a. Line 245: Please specify the days post PCR positive testing for COVID-19.....the time periods for acute and convalescent stages should be clearly indicated "during their acute disease and at long-term" is not specific enough

b. The "Median of days between symptoms onset and blood collection" is there a range to add to this? How was the 6 months time point determined for each patient.

11. Fig 8: Was the higher inflammatory signature for IL-6, IL-8, IP-10 evaluated as an index or integrated with the spectrum of IgA and other immune-cellular metrics? the current analysis for each molecule/marker individually is limited to direct two group comparisons PCC vs. non-PCC. The multi-profile of differences as a composite biomarker set of results would increase novelty for the PCC association.

12. A study limitations section is needed in the discussion.

a. The findings are highly applicable to hospitalized patients (given they represent 81% of the cohort)

b. Were Healthy controls tested for all parameters, serum antibodies? IgA? Confirmed negative for prior asymptomatic COVID-19 exposure?

13. Were there any repeated measures analyses included? It seems that changes over time in convalescents should be considered as a repeat measure on an individual from the baseline/acute sample...e.g. plasma cytokines

Minor concerns

- The words Pulmonary and Dyspnea are misspelled in Table 1. Please do thorough spell check.
- Line 120: hospitalized per diagnosis instead of due to COVID-19?
- A schematic diagram illustrating the immunological responses with the PCC would be helpful to follow the directions in each response, and to complete the data-driven story, hypothesis testing, and results interpretation. No hypothesis is currently explicitly stated.
- Was Ig antibody or IgA production tested from the supernatants of the PBMC stimulations?
- Line 464 "live" should be lived
- Please check grammar/sentences throughout (e.g. Line 485, line 488, line 502-505)
- Long COVID form is used in discussion, whereas PCC is used throughout intro-results sections, please list the multiple descriptions that will be used, or be consistent with one term.
- Line 524 conclusions: the statement infers that there is a spectrum of PCC severity. Does this exist for this cohort, and/or is this a study limitation if that spectrum of PCC symptom severity was not captured as part of this analysis?

Reviewer #3 (Remarks to the Author):

Santa Cruz et al. explore T cell phenotype and function in a cohort of COVID-19 convalescent patients presenting or not a post COVID-19 condition (PCC). A group of healthy individuals is used as control. The cohort includes hospitalized patients (with acute infection samples) and outpatients

(without retrospective sampling). The authors analyze CD8 T cell phenotype focusing on functional features, and retrospectively measure inflammatory responses to correlate with PCC. In addition, authors analyze alpha4 beta7 integrin expression (as a marker of mucosal homing) and anti-SARS-CoV-2 N and S IgA and IgG to assess potential viral persistence in mucosa. The differences observed in the latter parameters point to a potential role of viral persistence in PCC; however, there is no direct demonstration of viral presence in tissues.

The manuscript is well written, experimental approach appears to be sound and data is clearly presented; However, there are several issues that limit the relevance of the data presented:

General comments

Patient selection. The methods section describes selection criteria based on clinical data; however, the results section (lines 246-248) indicates that specific T or antibody responses were also used as selection criteria. Does this additional restriction affect patient selection? Some PCC individuals may show a non-seroconverter phenotype

Statistical analysis. It is unclear for this reviewer whether a correction for multiple comparisons has been performed.

Phenotype and function of SARS-CoV-2 specific T cells. Several features of CD8 T cells have been analyzed in the whole compartment. I would expect that main alterations would affect SARS-CoV-2 specific T cells. Could the authors comment on the phenotype of these cells?

Although the manuscript is focused on exploring potential causes of PCC by analyzing immune responses to SARS-CoV-2, it could be of interest to evaluate the potential diagnostic use of the identified T cell features. Have the authors explored this ??

Specific comments

The increased frequency of CD8 T cells 6 months after infection (Figure 1) is still controversial. How the authors explain this observation in fully recovered individuals?

T cell responses in HC (Figure 2). It is surprising that non COVID-19 individuals show a completely undetectable response (particularly to the N antigen). It has been widely demonstrated that some level of cross reactivity between SARS-CoV-2 and other human coronaviruses.

Besides T cell responses, IgA humoral responses are a relevant data for PCC. The authors report OD values in Figure 7. How these OD values were standardized among experiments?

Minor comments

Figure 7 is referenced as Figure 6 in lines 382-395

Response to Reviewers

Manuscript: NCOMMS-22-22439-T

Title: IgA to SARS-CoV-2 antigens and altered CD8⁺β7-integrin⁺ T cell characterized post COVID-19 condition

Dear Reviewers,

On behalf of all co-authors, I would like to express our gratitude for the comprehensive and constructive review, which contributed to improving the quality and focus of the manuscript. We also greatly appreciated the stimulating words of the referees throughout the review. The changes made to the manuscript were attentive to all these considerations and we feel that the final product is now of much greater scientific value.

All changes are highlighted in the revised version. We would like to point out that we have split the information previously presented on table 1 in two separate tables, which include the information related to the healthy controls demographics as requested by the three reviewers. We also decided to delete the old figures 1D-F, 5F and supplementary figure 4, where we performed sub-analysis on PCC and non-PCC groups divided by the severity during acute illness. We have also included a new co-author (Patrício Costa), a statistician, who provide considerable assistance in reviewing the statistical analysis performed to the revised manuscript. Finally, five new references were added.

Reviewer #1

- *This is a very nice paper assessing the immune response in convalescent individuals with PCC compared to convalescent asymptomatic and uninfected individuals. Some comments on statistical aspects of the paper:*

Major:

- *1. The authors mention on line 264 that the healthy controls were age- and gender-matched with the PCC and non-PCC cases. A description of the matching needs to be in the Patients and Study Design section of the paper.*
- *2. The statistical methods used should take into account the matched design of the study (they currently do not).*

In response to both queries. Indeed, the authors thank the three reviewers for pointing out this omission. We have changed Table 1 to include the information related to the healthy control's demographics and the appropriate statistical analysis. We have added a table 2 that includes the clinical characterization of the PCC and non-PCC cases during the hospitalization period, previously presented in table 1. Table 3 was also added, resuming the clinical parameters quantified during the consultation, six months after the infection.

Regarding the study design, healthy controls were selected by convenience, from patients with elective procedures that required a negative PCR test performed in the hospital. From the schedule list, we tried to get an average age of 60-65 years and a predominance of male patients. After checking the inclusion and exclusion criteria, patients were invited to participate. We recognize that this is not a perfect case-control matching procedure, so we have deleted the only erroneous reference to matching in the manuscript. However, we point out that no statistical differences were found between the three groups, as shown in the new table 1. The new table 1 includes all *p*-values and effect size measures, confirming the validity of this control sample. All statistics have been revised for this new version and are highlighted in yellow throughout the revised manuscript.

- *3. There are numerous statistical tests performed throughout the paper for the different outcome measures. Multiple testing corrections need to be used to control for the type-1 error rate.*

We acknowledge the incomplete description of the statistical methods. In fact, these corrections were performed using the Dunn's multiple comparisons test

upon all Kruskal-Wallis analysis. In addition, we performed a MANOVA using the quantified parameters used in our study. We found an overall significant difference with a Pillai's Trace value of 0.40, a $p < 0.001$ and a $\eta_p^2 = 0.98$ supporting our individual analysis for each variable. Two-way ANOVAs were performed on all variables considering the disease severity and the presence of post COVID condition as between-subject factors. Only the variables showing a p value less than 0.05 for the interaction between the two factors (severity of acute disease * development of PCC) were split according to these factors. Conversely, when this interaction was not observed, only PCC development was used as a factor among themselves. We clarify all these points in the Statistical Analysis section of Material and Methods (Lines 242-259).

- *Minor:*
- *Table 1: length of stay in hospital - please specify the unit of measure? I assume it is days? Also, length is misspelled.*

We apologize for what happened, in the new version of the manuscript we included the unit of measure (days).

Reviewer #2

- *“There is a need to understand physiological parameters and molecular signatures associated with long COVID. This manuscript highlights findings from an immunological profile analysis of adults with and without post-acute sequelae of COVID-19, referred to herein as post COVID condition (PCC). The peripheral blood mononuclear cells were examined for responsiveness to N and S antigens, as well as mucosal IgA levels, and plasma soluble factors/cytokines following SARS-CoV-2 infection from adults in both acute and convalescent stages. The overall goal was to explore immune cell marker indicators associated with development of post-COVID-19 conditions, such as fatigue and dyspnea. The exploratory nature of the study was clearly stated, but it seems some parameters and analysis was not fully disclosed. There was limited description regarding how PCC was established clinically across adults enrolled, and there did not seem to be any repeated measures statistical analysis completed for the two timepoints sampled (for any of the endpoint markers). There was additional lack of clarity on whether or not there were negative results, or lack of associations with immune markers, which is still important contributions to the emerging field of study. The findings for the persistent activated CD8 T cell subsets and higher levels of serum IgA was an interesting finding, but somewhat confusing to follow when discussed as compared to healthy controls. A major concern was that the healthy adult cohort demographics were not provided in Table 1, which makes this comparison a questionable control group. There were also some grammatical and spelling errors that merit attention during revision. A list of major and minor concerns are provided below to improve the overall clarity of the approach, results and conclusions.*

Major concerns

- *1. Line 105: “In this study, we explored the immune response associated with the persistence of symptoms upon acute SARS-CoV-2 infection”, re-write in Intro as a main objective, and provide a stated hypothesis? Were there other immunological measures explored that were omitted from these results? Understanding the immune parameters with negative/lack of association are also important to consider alongside the signature that is positive.*

We have stated the main objective and hypothesis of the work in the introduction section (Lines 100-103). In fact, in addition to the data presented in the manuscript, we evaluated other parameters on T lymphocytes and plasma from these individuals that failed to discriminate between HC and convalescent and even between non-PCC and PCC. Namely, the percentage of effector/naive and activation markers (CD28, CD95) on T cells or plasma cytokines such as TNF, IL-

1beta, IL-10, IL-12p40, IL-12p70 and GM-CSF. If the reviewer finds relevant, we may include these as supplementary data.

- 2. *Was the process for diagnosis/documentation of PCC collected in the same manner for each individual included in this cohort investigation? It seems that some PCC individuals could fall under two category types of PCC, thus have both fatigue and dyspnea. Please clarify this in cohort description.*

We acknowledge that in the Material and Methods section we did not clearly indicate that patients were diagnosed accordingly to the WHO criteria for Post Covid-19 condition, which were consistently applied to all our cohort. We have clarified this point in the new version of the revised manuscript (lines 164-165). Although we collected information on each patient's symptomatology (which allowed us to report the frequency of symptoms), we decided not to delve into this in detail for several reasons: (1) in clinical practice distinction between dyspnea and fatigue is not always clear and separation would make no difference in the diagnosis of the condition (both symptoms are included in the WHO definition of the syndrome [ref.25. Soriano, J. B., et al. A clinical case definition of post-COVID-19 condition by a Delphi consensus. *The Lancet Infectious Diseases* 3099, 19–24 (2021)], which is also characterized by clustering of symptoms; (2) by splitting patients in at least 3 subgroups (in either shortness of breath, fatigue, or both) would reduce statistical power, generate difficult interpretations, and not bring additional value.

- 3. *For the PCC in this cohort, were correlations measured between the IgA or CD8 T cell finding with these subsets of PC symptoms/conditions. This is relevant to understand the specificity of this immunological alteration to the myriad types of PCC reported in the literature. What falls under the category of "other" for the PCC.*

The issues raised by the reviewer are very pertinent. We should point out that the frequencies of symptoms in patients with PCC were described according the 5th domain of the consensus reached by the WHO: "Symptoms and/or impairments: cognitive dysfunction, fatigue, shortness of breath, others" [Soriano, J. B., et al. A clinical case definition of post-COVID-19 condition by a Delphi consensus. *The Lancet Infectious Diseases* 3099, 19–24 (2021)]. We should also mention that in our study, "others" represented anxiety (1), arthralgia (1), dysphonia (3), erectile dysfunction (1), hair loss (1), loss of appetite (1), muscle weakness (1), myalgia (3), palpitations (1), persistent cough (4), sadness (1), and thoracic pain (1), affecting 16 different patients. In order to make things clearer, we have transferred

the reported symptoms of PCC from the table to the text (Results - Demographic and clinical characterization of the cohort), including a detailed description of others as mentioned above (Lines 277-282).

- *4. Line 248-249. No demographics provided in Table 1 for the healthy controls (HC). Overweight should be added, with BMI cut off stated for obesity. Without these data on HC, the comparisons in the figures are difficult to understand for being matched or controlled for important variables such as age, sex, BMI, etc. The lack of HC demographics reduces enthusiasm for the rigor of the Data described across fig 4, 5, 6, 8*

As noted above, we acknowledge all three reviewers for pointing out this omission. We have amended Table 1 to include the information related to the healthy control's demographics and the appropriate statistical analysis. BMI was also included in Table 1. We have added a table 2 that includes the clinical characterization of the PCC and non-PCC cases, previously presented in table 1.

- *5. The Title does not fully depict the changes being measured, and title merits inclusion of the country cohort (Portuguese adults). Further, the term "altered" is unclear relative to the direction (increased?) markers of activation. a. Perhaps reword the finding to the title "Blood CD8+beta7 T cells and serum IgA humoral response against SARS-CoV-2 antigens characterized post COVID-19 condition at 6 months post infection in adults"*

We thank the reviewer for the suggestion. We have modified the title to "Blood CD8⁺β7⁺ T cells and serum IgA humoral response against SARS-CoV-2 antigens characterize Post COVID-19 Condition at six months post infection".

- *6. Were the patients/participants screened for SARS-CoV-2 positivity at enrollment and follow up visit (6 months)? The potential for re-infection was possible during these time periods and could be a factor with respect to the results reported at convalescent, and this should be clarified, if not verified. Please add the documentation for PCR/antigen positive test results to this to methods description for the acute phase as well as convalescent visits.*

In view of your comments, which we acknowledge, we have added the information in the Material and Methods section (Lines 118-119 and Lines 137-138). We did not screen the patients at the follow up consultation for SARS-CoV-2 due to the

absence of acute symptoms and clarified it (Lines 140-143). Thus, we agree that there is a risk of potential reinfection, although the incidence of the disease in the follow-up period was very low.

- *7. Did the stimulation of PBMCs from acute phase blood collection occur for N and S responsiveness, and test for differences between PCC and non-PCC? This would be a repeated measures comparison of acute and convalescent responses instead of only in convalescent stages. Given the large number of hospitalized patients included, these data are likely biased by many having moderate/severe acute disease conditions. Considering this and other issues raised below, we apologize if we have not been clear enough in our manuscript regarding the presented data. Therefore, we would like to make it clear that all data presented from figure 1 to 7 relate only to samples retrieved at 6 months post-COVID19, i.e. at SARS-CoV-2 follow up consultation. Only in figure 8, the data shown (levels of IL-6, IL-8 and IP-10) is from the same individuals but collected during acute infection. Of note, during the early pandemic, we did not perform PBMCs stimulation in the acute phase but only in samples collected at 6 months post-COVID19. Thus, repeated measures of T cells comparing the acute and the recovery phases were not applied here.*

- *Figure 1 and 2: the statistical comparisons in PBMC responses between PCC and Non-PCC, versus those from healthy controls are somewhat confusing in the results text and misleading. How was lymphopenia in the acute phase taken into consideration for influencing results?*

We apologize for not being clear. It is known that SARS-CoV-2 patients developed lymphopenia at distinct degrees of severity. We have indeed reported previously that T cells are more prone to die in the acute phase of infection and this may represent another confounding aspect of the development of Post-COVID-19 condition (Andre et al. CDD 2022). Whereas this will be of interest, this assay was not performed for this group of individuals. Yet, an analysis on the absolute lymphocyte count nadir of both sub-groups did not show any significant differences among them (shown in table 2).

- *a. Line 256-257 states “We did not observe any difference in PCC prevalence according to acute disease severity, need of hospitalization, length of stay, lymphocyte count or commonly used inflammation markers.”and unclear if this includes the PBMC stimulation response in acute stage?*

As mentioned above, we did not perform PBMC stimulation of samples collected during acute disease. Therefore, we cannot conclude if that may contribute to PCC prevalence.

- 8. *The finding that patients show distinct profiles in type I IFN, and mucosal type III compared to uninfected individuals supports existing publications. Please bolster support for this and reduce emphasis on this as a novel finding.*

- <https://www.science.org/doi/10.1126/science.abc6027>

- <https://www.nature.com/articles/s41467-021-27318-0> -

We do agree with the reviewer that a previous study, already cited in our manuscript [Phetsouphanh C, et al. Immunological dysfunction persists for 8 months following initial mild-to-moderate SARS-CoV-2 infection. *Nat Immunol* 23, 210–216 (2022)], have indicated higher levels of type I IFN (IFN- β) and type III IFN (IFN- λ 1) that remained persistently high at 8 months after infection. In our cohort, although we did not find any differences regarding IFN- λ 1 between PCC and non-PCC, an elevation on plasma IFN- β levels was observed in PCC. Previous studies, as the two mentioned papers above, have indeed demonstrated a possible role of type I and III IFN during COVID-19. Yet, both papers have addressed this issue during acute disease and not 6 months post-COVID19. The data presented in figure 5 demonstrate a higher IFN-beta and IFN-lambda2/3 levels are associated with PCC development in samples recovered at the follow-up appointment for SARS-CoV-2. We have clarified this issue in the discussion section (lines 543-548).

- *Given this, was a sample size calculation completed? This would add statistical rigor to the overall results.*

As previously mentioned, a MANOVA was conducted using all the cellular variables assessed in our study. The variable that described the PCC symptoms was applied as a fixed factor. We found significant differences, with a Pillai's Trace value of 0.40, a $p < 0.0010$ and $\eta_p^2 = 0.98$. The a priori calculation of the sample size is challenging for this type of study since we cannot forecast how many patients we will be able to enroll. Thus, we decided to determine the achieved statistical power, considering the results of the MANOVA test. Using G*Power software, the achieved statistical power was computed as a Post hoc analysis given α , sample size and effect size. The effect size was calculated based on Pillai Value of 0.444, three groups and twelve variables (cellular characteristics evaluated in our study). Considering the estimated effect size (0.25), an α of 0.05, a sample size of 164

patients, three groups and twelve responsive variables were used, the achieved power of our study is 0.99.

- 9. *In methods “At admission and during hospitalization, daily information on the disease stage, need of respiratory support, treatments used, diagnosis of pulmonary embolism if present, tomography data, blood count, quantification of C-reactive protein (CRP), ferritin, d-dimer, and procalcitonin.” they are missing from the 6 month follow up analysis.*

The daily clinical variables of acute disease were used to identify the worst day of the disease and to summarize the severity of patients, as it could be related to PCC. These data and basic laboratory parameters are presented in table 2. Table 3 describes the laboratory parameters assessed during the consultation six months after infection including lymphocyte counts and the C-reactive protein (CRP), ferritin, and d-Dimer quantifications in both Non-PCC and PCC subjects.

- a. *What sort of follow up PCC vs Non-PCC analysis of these key clinical parameters was performed? Notably, these analytes also merit further explanation in the post COVID condition for associations to the recovery phase, alongside immunological findings (IgA, CD8 signature etc.)?*

We fully agree on the importance of accurate follow-up. What we have done in terms of a follow-up analysis can be seen in tables 2 and 3. In table 2 we compare the clinical and laboratory parameters of acute disease in PCC vs Non-PCC patients. In table 3, we compare the laboratory parameters of both groups during follow-up, showing no differences. No other positive or negative correlations with immunological results were found. In this sense our data reinforce the idea that the most commonly used clinical and laboratory parameters are not of much interest in explaining, diagnosing, or predicting post-COVID conditions.

- 10. *Eligibility criteria: were all patients required to provide a PCR or antigen test positive report, and if yes, these would have dates to indicate time frame of symptoms/active infection, but that information is limited in this report. Is acute phase within days 1-15 from test positive result or symptoms?*

We appreciate this observation. It is true that all patients were required to provide a positive PCR report or antigen test. We clarified this information in the revised manuscript: patients were only diagnosed and included in the presence of COVID-19 symptoms and a positive PCR test (lines 118-119 and lines 137-138). We define the acute disease starting point based on symptomatology.

a. Line 245: Please specify the days post PCR positive testing for COVID-19.....the time periods for acute and convalescent stages should be clearly indicated “during their acute disease and at long-term” is not specific enough

b. The “Median of days between symptoms onset and blood collection” is there a range to add to this? How was the 6 months time point determined for each patient.

We apologize for not being clear. As required, we present in table 2 the period between symptoms and diagnostic, hospitalization, and appointment with the respective comparison between the two groups. No differences were found between groups.

11. Fig 8: Was the higher inflammatory signature for IL-6, IL-8, IP-10 evaluated as an index or integrated with the spectrum of IgA and other immune-cellular metrics? the current analysis for each molecule/marker individually is limited to direct two group comparisons PCC vs. non-PCC. The multi-profile of differences as a composite biomarker set of results would increase novelty for the PCC association.

The aim of the cytokine profile during hospitalization was to assess a model to predict the occurrence of PCC symptoms six months post infection. However, when we conducted a binary logistic regression, using the three cytokines quantified during the acute phase of the disease (IL-8, IL-6 and IP-10), we obtained a non-significant predictive model with $\chi^2=1.742$, $df=3$, $p=0.628$ and a Nagelkerke R^2 of 0.033. Thus, the data reported in figure 8 suggests a much higher level of inflammation during hospitalization in PCC patients compared to non-PCC patients and healthy controls.

• 12. A study limitations section is needed in the discussion.

a. The findings are highly applicable to hospitalized patients (given they represent 81% of the cohort)

A study limitations section was included in the discussion with mention of the high proportion of hospitalized patients in our cohort (Lines 590-600).

b. Were Healthy controls tested for all parameters, serum antibodies? IgA? Confirmed negative for prior asymptomatic COVID-19 exposure?

Yes. All healthy controls used in our study were tested for all parameters, including the presence of antibodies to SARS-CoV-2 antigens. From our initial group of 39 Healthy controls, we excluded 2 individuals considering their positive

reactivity for the presence of anti-S and anti-N IgG. We have included a sentence in the Patients and Study Design section to address this issue (Lines 149-150).

- 13. *Were there any repeated measures analyses included? It seems that changes over time in convalescents should be considered as a repeat measure on an individual from the baseline/acute sample...e.g. plasma cytokines.*

We recognize that our manuscript did not make sufficiently clear that we do not analyze the same parameters longitudinally during hospitalization on the same individuals. Although we collected blood samples at admission and every 72 h during hospitalization, for this study we used a sample corresponding to the worst clinical timepoint, defined as the highest respiratory support. Thus, all data shown in figures 1 to 7 relate only to samples recovered at 6 months post-COVID-19, i.e. at the SARS-CoV-2 follow up consultation. Only in figure 8, we show data from samples collected during acute infection. So, we have clarified this issue in the resubmitted manuscript in the Patients and Study Design section, accordingly, making it much clearer (lines 122-125 and 131-139).

- *Minor concerns*

- *The words Pulmonary and Dyspnea are misspelled in Table 1. Please do thorough spell check.*

We have corrected the misspelled words in Table 1 and checked the grammar and spelling throughout the manuscript.

- *Line 120: hospitalized per diagnosis instead of due to COVID-19?*

The sentence was corrected (Line 118).

- *A schematic diagram illustrating the immunological responses with the PCC would be helpful to follow the directions in each response, and to complete the data-driven story, hypothesis testing, and results interpretation. No hypothesis is currently explicitly stated.*

A schematic diagram was added to the revised version of the manuscript.

- *Was Ig antibody or IgA production tested from the supernatants of the PBMC stimulations?*

No, we have not quantified any Ig production on the PBMC supernatant cultures.

- *Line 464 “live” should be lived*

It was corrected.

- *Please check grammar/sentences throughout (e.g. Line 485, line 488, line 502-505)*

It was corrected.

- *Long COVID form is used in discussion, whereas PCC is used throughout intro-results sections, please list the multiple descriptions that will be used, or be consistent with one term.*

Throughout the text, the “Long Covid form” expression was replaced by Post-Covid Condition (PCC).

- *Line 524 conclusions: the statement infers that there is a spectrum of PCC severity. Does this exist for this cohort, and/or is this a study limitation if that spectrum of PCC symptom severity was not captured as part of this analysis?*

The sentence is not correct. We did not sub-analyse the PCC patients regarding the severity of their condition. The sentence was corrected in the revised manuscript (lines 600-603).

- **Reviewer #3**

- *Santa Cruz et al. explore T cell phenotype and function in a cohort of COVID-19 convalescent patients presenting or not a post COVID-19 condition (PCC). A group of healthy individuals is used as control. The cohort includes hospitalized patients (with acute infection samples) and outpatients (without retrospective sampling). The authors analyze CD8 T cell phenotype focusing on functional features, and retrospectively measure inflammatory responses to correlate with PCC. In addition, authors analyze alpha4 beta7 integrin expression (as a marker of mucosal homing) and anti-SARS-CoV-2 N and S IgA and IgG to assess potential viral persistence in mucosa. The differences observed in the latter parameters point to a potential role of viral persistence in PCC; however, there is no direct demonstration of viral presence in tissues.*

- *The manuscript is well written, experimental approach appears to be sound and data is clearly presented; However, there are several issues that limit the relevance of the data presented:*

- *General comments*

- *Patient selection. The methods section describes selection criteria based on clinical data; however, the results section (lines 246-248) indicates that specific T or antibody responses were also used as selection criteria. Does this additional restriction affect patient selection? Some PCC individuals may show a non-seroconverter phenotype*

Specific T or antibody responses were used only in the control group to exclude potential individuals who had previously been unaware of contact with SARS-CoV-2. Two individuals were excluded due to positive reaction as noted in the Materials and Methods section. We had re-written the original sentence for clarity (Lines 149-150 and 266-267).

- *Statistical analysis. It is unclear for this reviewer whether a correction for multiple comparisons has been performed.*

We acknowledge the incomplete description of the statistical methods. As replied to reviewer 1, these corrections were performed using Dunn's multiple comparisons test upon all Kruskal-Wallis analysis. We have added this information in the revised Material and Methods section (Lines 242-259).

- *Phenotype and function of SARS-CoV-2 specific T cells. Several features of CD8 T cells have been analyzed in the whole compartment. I would expect that main alterations would affect SARS-CoV-2 specific T cells. Could the authors comment on the phenotype of these cells?*

We agree with the reviewer that specific T cells could be impacted. However, the average magnitude of the detected SARS-CoV-2-specific CD8 T cell responses is estimated in 1.7% (range: 0.005–19%) of the total CD8+ T cells [Gangaev, A., et al. Identification and characterization of a SARS-CoV-2 specific CD8+ T cell response with immunodominant features. *Nat Commun* 12, 2593 (2021)]. Thus, this percentage cannot explain the observed changes that were observed in the total CD8 population that is higher. As an example, the mean percentage of CD8+GzmB+ was of 55.10% for HC, 63.74% for Non-PCC and 67.17%, an increase of 8.64% and 12.07% for non-PCC and PCC, respectively, compared to HC that could be only explained by the SARS-CoV-2-specific CD8 T cells. Either the estimation of specific SARS-CoV-2 CD8 T cells, based on the assay used, is lower or most probably our results reflect bystander effect on CD8 T cells. Indeed, the induction of bystander CD8 T cells with effector functions and their role in immunity to infections, particularly other viral infections, has been described (Reviewed by Tae-Shin Kim and Eui-Cheol Shin, *Exp. Mol. Med.*, 2019). We address this issue in the discussion (Lines 552-557).

- *Although the manuscript is focused on exploring potential causes of PCC by analyzing immune responses to SARS-CoV-2, it could be of interest to evaluate the potential diagnostic use of the identified T cell features. Have the authors explored this ??*

The authors agree with the reviewer. Indeed, not only the T cell features may be of interest as a potential diagnostic tool but also, and perhaps more importantly, the anti-N and anti-S IgA response. Thus, looking at the anti-S and anti-N IgA data, we detected 88% of positivity for anti-N IgA in PCC individuals when compared to only 40% in non-PCC convalescents. Moreover, 57% of PCC produced both anti-N and anti-S IgA when compared to only 12.9% of non-PCC. Therefore, we believe that the potential use for anti-N and anti-S IgA as a diagnostic tool should be validated in larger cohorts.

- *Specific comments*

- *The increased frequency of CD8 T cells 6 months after infection (Figure 1) is still controversial. How the authors explain this observation in fully recovered individuals?*

We thank the reviewer for pointing out this issue. Indeed, as shown in the supplementary figure 1 and 2, we quantified both CD4 and CD8 populations within the T lymphocyte population as defined by CD3 positivity. Therefore, we should not assume increase numbers of CD8 T cells, but instead increased relative percentage of CD8 T cells within the CD3 population. We modify the text accordingly (lines 289-290 and 292). The median of CD3+CD4+ population was 59.0%, 53.2% and 54.7% for HC, non-PCC, and PCC, respectively, without any statistical difference. While the median of CD3+CD8+ population was 34.6%, 45.0% and 43.9% for HC, non-PCC and PCC, respectively. Yet, after our initial submission and during the peer-reviewing, other publications have reported increased CD8 T cells during convalescence, although without discriminating between non-PCC and PCC (Govender et al, *Frontiers in Immunology*, 2022 and Lyudovyyk et al, *Cancer Cell*, 2022).

- *T cell responses in HC (Figure 2). It is surprising that non COVID-19 individuals show a completely undetectable response (particularly to the N antigen). It has been widely demonstrated that some level of cross reactivity between SARS-CoV-2 and other human coronaviruses.*

We agree with the reviewer that some level of cross-reactivity with other human coronaviruses was initially reported. However, recent data, as observed in other recent publications such as Nelson et al, *Sci. Immunol*, 2022 and Villemonteix et al, *Imm Inflamm Dis*, 2022, have shown the absence of reactivity of healthy controls to SARS-CoV-2 antigens similarly to what is reported in our manuscript.

- *Besides T cell responses, IgA humoral responses are a relevant data for PCC. The authors report OD values in Figure 7. How these OD values were standardized among experiments?*

Quantification of the humoral response is based on assays we have developed to accurately quantify the Ig response in patients with COVID-19. This quantification was recently reported in our manuscript published in *Cell Death and Disease* (André S et al, *CDD*, 2022). As you will see, the assay is standardized, and serial plasma dilution is performed to determine non-specific recognition. Thus, plasma from HD is tested concomitantly with plasma from patients. Thus, the values were established on diluted HD plasma, and values shown are OD values of at 1/400

dilution for both IgA and IgG. For accurate quantification, positive references are included for standardization.

- *Minor comments*

- *Figure 7 is referenced as Figure 6 in lines 382-395*

It was corrected.

REVIEWER COMMENTS

Reviewer #1 (Remarks to the Author):

The authors have adequately addressed my previous concerns.

Reviewer #2 (Remarks to the Author):

A number of revisions that have been made improved the clarity of the overall findings for the role of the mucosal immune system. The healthy controls are better described alongside the main findings of IgA, CD8 T cell persistence/activation in PCC. A few comments remain regarding the revised version.

1. Line 299-300, thanks for clarification that there is no relationship/interaction between CD4, CD8, or ratios with the acute disease severity and PCC. Notably, all convalescent results are shown in Fig 1-7, but then in Figure 8 is the switch to describing the acute condition. There remains some nuance to this understanding because this data shows results from the patients across PCC groups in acute stage...and the data demonstrates increased levels of inflammatory mediators in plasma of acute SARS-CoV-2 patients that later developed PCC.

Showing acute and convalescent plasma data together could indicate stronger linkage between the inflammation at baseline and the persistence with PCC. It is not clear why both timepoints are not presented in Figure 8, which regardless of finding will enhance the results interpretations.

The IL6/IL8/IP10 were not compared to levels of these soluble cytokines at 6 months. Why?

The repeated measures does seem possible for plasma parameters in Figure 8, and thus for other plasma markers presented in paper (e.g. IgA, Table 3 data for d-dimer, CRP).

2. The lack of IgA and CD8 T cell data during acute infection (and for comparison to 6 months post infection) is a limitation that is not included in the discussion paragraph with other study limitations.

3. Are there any viral load-titer data available from acute phase or from the hospital? did any patients report knowledge of possible repeat infection within the 6 months? these are other factors to consider that may support the speculation of viral persistence in mucosa.

Reviewer #3 (Remarks to the Author):

The authors have improved the manuscript by including better descriptions of the cohort and the statistical methods.

All questions have been addressed

Response to Reviewers

Manuscript: NCOMMS-22-22439A

Title: Blood CD8+ β 7+ T cells and serum IgA humoral response against SARS-CoV-2 antigens characterize Post COVID-19 condition at six months post infection

Dear Reviewers,

On behalf of all co-authors, I would like to express our gratitude for the positive feedback provided by reviewers #1 and #3. We also acknowledge the further queries by reviewer #2. The changes made to the manuscript were attentive to these considerations and are highlighted in the revised version. We would like to point out that we added one reference (ref. 121), one supplementary figure and one supplementary table.

Reviewer #2

Reviewer #2 (Remarks to the Author):

- *A number of revisions that have been made improved the clarity of the overall findings for the role of the mucosal immune system. The healthy controls are better described alongside the main findings of IgA, CD8 T cell persistence/activation in PCC. A few comments remain regarding the revised version.*

- *1. Line 299-300, thanks for clarification that there is no relationship/interaction between CD4, CD8, or ratios with the acute disease severity and PCC. Notably, all convalescent results are shown in Fig 1-7, but then in Figure 8 is the switch to describing the acute condition. There remains some nuance to this understanding because this data shows results from the patients across PCC groups in acute stage...and the data demonstrates increased levels of inflammatory mediators in plasma of acute SARS-CoV-2 patients that later developed PCC. Showing acute and convalescent plasma data together could indicate stronger linkage between the inflammation at baseline and the persistence with PCC. It is not clear why both timepoints are not presented in Figure 8, which regardless of finding will enhance the results interpretations. The IL6/IL8/IP10 were not compared to levels of these soluble cytokines at 6 months. Why? The repeated measures does seem possible for plasma parameters in Figure 8, and thus for other plasma markers presented in paper (e.g. IgA, Table 3 data for d-dimer, CRP).*

We chose not to include the requested data on our previous versions of the manuscript given that the findings would not, in our perspective, enhance the

interpretation of the mechanisms involved in the PCC development. Yet, at your request, we have added one supplementary figure showing the levels of IL-6, IL-8, IP-10, CRP, D-Dimer and Ferritin during acute disease and at convalescence divided by the individuals that did not develop PCC and those who did (Supplementary figure 6). A repeated measures analysis of all variables comparing acute with convalescence samples did not show any statistical significance. We have clarified these issues on the manuscript on lines 546-550.

- 2. *The lack of IgA and CD8 T cell data during acute infection (and for comparison to 6 months post infection) is a limitation that is not included in the discussion paragraph with other study limitations.*

We have added a sentence on the discussion section, lines 597-601.

- 3. *Are there any viral load-titer data available from acute phase or from the hospital? did any patients report knowledge of possible repeat infection within the 6 months? these are other factors to consider that may support the speculation of viral persistence in mucosa.*

We acknowledge that this is a relevant question. Both viral load at acute disease and potential repeat infection could contribute to viral persistence evidence in mucosa or even PCC development.

The quantification of the viral load was not initially performed by routine at the hospital. Some patients arrived at the hospital already with a positive PCR test performed by licensed private laboratories, whose data is not available to us. Some other patients performed the test at the hospital when their condition was milder, days before the hospitalization, which could be a confounding factor for the analysis. Thus, systematic quantification of viral load at admission was not always possible. While detection of SARS-CoV-2 in the blood of convalescent patients is not relevant, analyses of stool could be an alternative. However, for safeguarding patient comfort and to guarantee easier patient's recruitment, acceptance, and compliance to the study, this was not proposed. Moreover, in the clinical interview patients were asked about any COVID-19 related symptomatology between the discharge and the consultation. None have communicated any suspicion of repeat infection.

Although we did not demonstrate the presence of the virus in the mucosa, the persistence of SARS-CoV-2 particularly in the gastrointestinal system has been documented and may remain for more than several months after acute infection. These publications have already been cited on our manuscript.

- Wu, Y. et al. Prolonged presence of SARS-CoV-2 viral RNA in faecal samples. *Lancet Gastroenterol Hepatol* 5, 434–435 (2020);
- Sun, J. et al. Prolonged persistence of SARS-CoV-2 RNA in body fluids. *Emerg Infect Dis* 26, 1834–1838 (2020);
- Vibholm, L. K. et al. SARS-CoV-2 persistence is associated with antigen-specific CD8 T-cell responses. *EBioMedicine* 64, 103230 (2021);

- **Gaebler, C. et al. Evolution of antibody immunity to SARS-CoV-2. Nature 591, 639–644 (2021).**

REVIEWERS' COMMENTS

Reviewer #1 (Remarks to the Author):

I believe the authors have adequately responded to Reviewer 2's additional comments.